Guidelines for a participatory Smart City model to address Amazon’s urban environmental problems

http://orcid.org/0000-0002-1718-7989 da Silva Jonas Gomes jgsilva@ufam.edu.br
Industry Engineering Department-Eureka Laboratory, Universidade Federal do Amazonas , Manaus, Amazonas , Brazil
Rosak-Szyrocka Joanna
Electronic publication date: 2023 Dec 12
Publication date: 2023
Volume: 9
Electronic Location ID: e1694
Received 2023 Apr 19; Accepted 2023 Oct 20
Copyright: © 2023 da Silva
Copyright year: 2023
Copyright holder: da Silva
License: This is an open access article distributed under the terms of the Creative Commons Attribution License, which permits unrestricted use, distribution, reproduction and adaptation in any medium and for any purpose provided that it is properly attributed. For attribution, the original author(s), title, publication source (PeerJ Computer Science) and either DOI or URL of the article must be cited.
License URL: https://creativecommons.org/licenses/by/4.0/

Keywords: Citizens engagement, Climate change, CO2, Data, Decarbonization, Digital technologies, Disruptive technologies, Environment, Smart city, Sustainable city

Funding: This research received no external funding support, with all required investments provided through the author’s personal funds.

==============================
Climate change is a global challenge, and the Brazilian Amazon Forest is a particular concern due to the possibility of reaching a tipping point that could amplify environmental crises. Despite many studies on the Amazon Forest, this research was conducted in Manaus, the capital of Amazonas state, to address five gaps, including the lack of local citizen consultation on urban environmental issues, Smart Cities, decarbonization, and disruptive technologies. This study holds significance for the academy community, government bodies, policymakers, and investors, as it offers novel insights into the Amazon region and proposes a model to engage citizens in Smart Cities. This model could also guide other municipalities aspiring for participatory sustainable development with a decarbonization focus, mitigating future risks, and protecting future generations. Basically, it is an explanatory and applied study that employs mixed methods, including literature, bibliometric and documentary reviews, two questionnaires, and descriptive statistical approaches, organized in four phases to reach the following goals: (a) provide information on the main challenges facing humanity, the Brazilian Amazon state, and the city of Manaus; (b) identify the best Smart City approaches for engaging citizens in solving urban problems; (c) contextualize and consult Manaus City Hall about the effectiveness of the Smart City project; (d) investigate the perceptions of citizens living in Manaus on the main city’s environmental problems, as well as their level of knowledge and interest on issues related to Smart Cities, decarbonization, and disruptive technologies; (e) propose a participatory Smart City model with recommendations. Among the result, the study found that the term “Smart City” dominates scholarly publications among nineteen urban-related terms, and the five main environmental problems in Manaus are an increase in stream pollution, garbage accumulation, insufficient urban afforestation, air pollution, and traffic congestion. Although citizens are willing to help, the majority lack knowledge on Smart City and Decarbonized City issues, but there is a considerable interest in training related to these issues, as well as disruptive technologies. It was found that Amsterdam, Melbourne, Montreal, San Francisco, Seoul, and Taipei all have a formal model to engage citizens in solving their urban problems. The main conclusion is that, after 6 years, the Smart City Project in Manaus is a political fallacy, as no model, especially with a citizen participatory approach, has been effectively adopted. In addition, after conducting a literature and documentary review and analyzing 25 benchmark Smart Cities, the P5 model and the Citizen Engagement Kit model are proposed with 120 approaches and guidelines for addressing the main environmental problems by including Manaus’ citizens in the Smart City and/or decarbonization journey.

Introduction

Climate change is a global challenge, a long-term change in weather and temperature patterns with intense consequences for humanity and the planet (UN, 2020). Moreover, climate and environmental risks are the core focus of global risk perceptions over the next decade, but they are risks we are seen to be the least prepared for (World Economic Forum, 2023, p. 8).

Climate change publications date back to the early 1900s (De Courcy Ward, 1906a, 1906b; Lockyer, 1910; Humphreys, 1913; Nature, 1913; Agassiz, 1938), but it has become one of the main scientific concerns, since the 1992 Rio-92 (Fig. 1), with 328,000 publications (Articles, Chapters, Proceedings, and Edited Books) with titles or abstracts related to these words (Dimensions, 2022), with average growth per year equal to 16.6%. In addition, for every 5 years, since 1992, the total number of publications is growing, reaching the highest growth (237%) in the fourth quinquennium (Q4 = 2007 to 2011 = 42,333 publications) compared with Q3 (12,578 publications).

Figure 1 Number of publications on climate change since 1992.

Population growth is a growing concern. According to the UN (2019b), it is estimated that by 2050, 68% of the global population will reside in urban areas. While cities are contributors to climate change, they also play crucial roles in its mitigation and prevention. Therefore, if we consider population growth, climate change risks, and effects in the future (IPCC, 2022), while city areas occupy around 2% of the world’s land, consume 2/3 of its energy, and emit 75% of its carbon emissions (The World Bank, 2021), it is possible to imagine that urban life could be difficult in the future (Yukiko, 2021) if we do not address the main environmental challenges towards the Low Carbon Economy transitions.

One way some nations/cities are making the transition is through Smart City policies, since they can improve their low-carbon economy by integrating new technologies into the cities’ operation and management, optimizing the energy’s supply and demand, as well as information sharing among government, enterprises, and citizens (Fan, Peng & Liu, 2021; Gomes da Silva, 2022a).

For conciseness, the frequently used terminology of “Smart City or Smart Cities”, “Disruptive Technologies”, and “Citizens Engagement” will be abbreviated as SC, DT, and CE hereafter.

Research’s main context, gaps, and originality

It was realized in Brazil, the fourth largest global CO2 emitter since 1850 (Evans, 2021). Most emissions are from land use and forests, highlighting that 2020’s Brazilian Amazon deforestation rate was the highest of the decade (Silva Junior et al., 2020) and emitted 20% more CO2 into the atmosphere than it absorbed in 10 years (Qin et al., 2021).

Although several studies developed in the Brazilian Amazon region focused on the forest, no study has been conducted in Manaus’ urban area, taking into consideration local citizens’ perceptions of environmental, SC, decarbonization, and DT issues. This gap cannot be ignored by the academy and global society, because Manaus is 352 years old and:

(a) has 2.063 m citizens living in an area of 11,401 Km2 (IBGE, 2021a), which represents almost the size of Qatar, Jamaica, or Lebanon. It is the capital and the 35th largest city of the Amazon State, considered the biggest state in Brazil (IBGE, 2021b).

(b) its GDP in 2019 was considered the sixth largest of Brazil (IBGE, 2021c, p. 3) due to Industrial Park, which in 2021 alone earned the amount of R$158.62 bi (Suframa, 2022a), through the performance of companies such as Honda, P&G, LG, Samsung, Sony, etc.

(c) the city, from 2000 until 2018, emitted around 10.7 MCO2 (70.3% to produce energy), which represented almost 10% of the Amazon State’s CO2 emission (SEEG Brasil, 2022).

(d) according to JICA (2010), for 18 months, 187 Manaus Industrial Park companies produced 628.9 tons of waste/day, of which 120 tons were considered hazardous industrial waste.

(e) it is suffering from pollution, floods, fires, health problems, violence, and social disparities. As a city that emerges from the forest, it is important to think about sustainable urban planning solutions that can address the urban environmental problems from the citizen’s perspective (de Medeiros, da Fonseca & de Silva, 2020).

(f) the most intriguing fact, in 2016, is that its mayor, Arthur Virgílio Neto, was reelected, after making a strong electoral campaign promising a project to transform Manaus into a SC, but the marketing campaign ceased once he was re-elected.

Although several authors (Tan, 1998; Mahizhnan, 1999; Giffinger et al., 2007; Dameri, 2013; Capra, 2014; Albino, Berardi & Dangelico, 2015; Fernandez-Anez, 2016; Eremia, Toma & Sanduleac, 2017; Fernandez-Anez, Fernández-Güell & Giffinger, 2018; Santos et al., 2018; IAP2 International Federation, 2018; Lai et al., 2020; Janik, Ryszko & Szafraniec, 2020; Belausteguigoitia et al., 2022; Jiang, Geertman & Witte, 2022; Puron-Cid & Gil-Garcia, 2022; UN-Habitat, 2022 etc.) have published valuable studies on SC and/or models to engage citizens, there are gaps that need to be addressed to provide updated information about:

(Gap2) the main terms used over time to address urban challenges; (Gap3) the key publications, facts, and enablers that contributed to the evolution and popularization of the term “Smart City”; (Gap4) enhancing understanding of SC foundations, related to its enablers, definition, type of city (Vision), digital technologies used with approaches to engage citizens; (Gap5) the development of a model to transform Manaus’ citizens into protagonist participants during the SC journey.

The study is original and differs from others in its geographic focus on citizens perceptions in Manaus, filling a gap, as most previous Brazilian SC publications centered on cities or issues in the country’s south or south-central regions (Gomes da Silva, 2023).

Additionally, the interdisciplinary approach utilizes mixed methods, exploring historical and practical perspectives. The extensive research fills gaps through a long-term analysis in five scientific databases, spanning 122 years, followed by real cases learned from 25 of the world’s best SC, resulting in the development of a practical, flexible, and participatory SC Model for Manaus’s policymakers, public managers, and others.

Main questions, goals, importance, social impacts, and implications

(Q1) given climate change and population growth, how can we protect future generations? (Q2) Does Manaus City Hall SC Project work? (Q3) How do Manaus’ residents view SC, decarbonization, DT, and urban environmental issues? (Q4) which SC have the most inspiring citizen engagement models? (Q5) How can Manaus’ challenges be addressed using a citizen-centric SC model?

Goals: (a) provide information on the main challenges facing humanity, the Brazilian Amazon state, and Manaus; (b) identify the best SC approaches for engaging citizens in solving urban problems; (c) contextualize and consult Manaus City Hall about the SC project’s effectiveness; (d) investigate the perceptions of Manaus citizens regarding the main city’s environmental problems, as well as their level of knowledge and interest on issues related to SC, Decarbonization, and DT; (e) propose a participatory SC Model with recommendations to local managers.

Importance: it has multifaceted social impacts and implications for academia, policymakers, investors, authorities, and practitioners in the fields of CE, public administration, urban planning, and sustainability.

For the academy, the research provides new insights, offers scholars an in-depth understanding of citizens’ perceptions regarding SC, decarbonization, and DT in an Amazonian urban setting. This can contribute to a more geographically diverse understanding of these subjects. It also contributes to the SC education process, and its recommendations open opportunities for new studies. As a result, in the medium and long term, behavior changes can be made, with increased understanding and involvement, residents might change their behaviors in favor of more sustainable practices and participate more actively in SC initiatives.

It is important research for public authorities who wish to correctly start the SC journey, especially those working at Manaus City Hall, since they will gain valuable insights from the best SC background, and from Manaus’ residents. By identifying best CE practices, policymakers can learn and make policy changes, developing strategies to involve residents more fully in the decision-making process, thereby enhancing the legitimacy and effectiveness of their policies, strengthen democratic process, foster a sense of belonging and community, and encourage citizens to play a more active role in shaping their city.

It can be useful for investors and authorities interested in a better understanding of the Amazon State and Manaus city, since the region is still unknown to many people living outside of Brazil.

In terms of sustainability, the study addresses climate change, one of the most pressing issues facing humanity. By exploring approaches to engage inhabitants in identifying the main environmental issues, decarbonization and DT, this paper could contribute to mitigation efforts and help safeguard the wellbeing of future generations.

Finally, it can scale solutions because the lessons learned from the best SC and the proposed participatory SC model could serve as a guide for other cities seeking to promote participatory sustainable development, potentially transforming urban life on a larger scale.

Survey methodology

It is an explanatory and applied study that employs mixed methods, including literature, bibliometric and documentary reviews, two questionnaires, and descriptive statistical approaches. The methodology consists of four phases to gather data and analyze it in a comprehensive manner:

Phase 1: general literature, bibliometric, and documentary review

It is based on general literature research, the study of articles, books, policies, guidelines, manuals, official sites, government programs, decrees, standards, technical reports, dissertations, and theses, collected from the internet. In terms of bibliometric studies, investigations were realized between 19th July 2022 and 25th August 2022, in five scientific databases (Lens.org, Dimensions.ai, Engineering Village, Web of Science, and Science Direct), to find publications from the 1900s until 2022, good cases of SC and other information, to support this research.

Concerning to documentary research, it focused on practical cases, and it was developed between September 2022 and March 2023, by consulting the official sites of specialized organizations or governments considered Benchmark SC, and official sites related to Manaus City Hall.

Phase 2: contextualization, and the first diagnosis, consulting the City Hall Managers

The contextualization was based on a review of public documents published between 2016 and 2022, such as the government plan approved by the Brazilian Superior Electoral Court for the election of mayor in 2016 (Coligação por uma só Manaus, 2016) and in 2020 (Coligação Avante Manaus, 2020), as well as decrees published by the Official Gazette of Manaus City, to identify the main decisions related to Manaus SC Project.

A questionnaire with seventeen open questions (Appendix 1), based on factors used by Eden Strategy Institute (2021) was submitted to the Manaus City Hall Managers on May 23, 2022, via the city’s transparency portal (https://transparencia.manaus.am.gov.br/transparencia/v2/#/lai), protocol 2831/2022. This is the main channel for residents to ask and receive public information according to the Law No. 12527 and Decree No. 4157. Note that no respondent identification information was required.

Phase 3: realize the second diagnosis, consulting Manaus citizens

On December 8, 2021, using Typeform, an electronic questionnaire was created in the Portuguese language (https://quiz.typeform.com/to/XX7dbMg4), available in English (Appendix 2), with a welcome, target audience (>=18 years old), goals, eleven multiple-choice questions, and two open questions.

In terms of sample size, the Brazilian Institute of Geography and Statistics (IBGE) is realizing a new census in Brazil, and from the last census (IBGE, 2010), it is possible to estimate that, in that time, around 66% of the Manaus population was aged 18 or over. The current census is over, and the IBGE’s last estimate of Manaus’ population was 2,063,547 people in 2022 (IBGE, 2021a). If the age group proportion has not changed, 66% is 1,361,941.02 individuals.

Considering the values of 95% confident level, 3% margin of error, 50% response distribution, and a population target of 1,500,000 people, a representative sample of 1,067 respondents was estimated (SurveyMonkey, 2022; Raosoft, 2022). However, due to the possibility of receiving a questionnaire with incomplete data or other issues, a target of at least 1,300 respondents was set.

The questionnaire was pilot-tested from 8–15 December 2021 to assess its comprehensiveness, and two improvements were made to make it easier. After that, the survey ran from 16 December 2021 to 9 December 2022, with 1,308 respondents, 1,242 of whom were correct and 66 eliminated due to missing, repeated, or under-18 data.

To invite people, the following strategies were used:

Strategy (1) when the author spoke on a SC public policy panel at the Third Fair of Manaus Digital Pole on December 9, 2021. Strategy (2) in Amazon Federal University lectures. Strategy (3) by establishing a Facebook invitation to the questionnaire and posting it in local Manaus groups (universities and neighborhoods). Strategy (4) using Facebook’s “Boost post” tool to invite 105,939 Manaus residents in 95 days for R$1505.62.

Due to COVID-19 and cost limitations, interviews were not possible, and the researcher continued the data collection and verification in 2022 to reach the target goal, which depended on the availability of the respondent to voluntarily answer the survey.

Phase 4: propose a participatory SC model with recommendations

The model is based on practical cases, especially from publications identified in phase 1, and 25 Benchmark SC selected from the list of five International ranking specialized reports published in the last 2 years by the Eden Strategy Institute (2021 p. 2), IMD SC Observatory and SUTD (2022), IESE Business School (2022, p.26), and DTTM, ISi Lab and IfM Engage (2022), and The Economist Group (2022, p. 49).

The main criteria to select each ranking are to (1) have been published since 2021; (2) be related to Smart or Digital City; (3) have an international list of at least 30 cities; (4) have free access to the report or the list of the best cities.

To select the best SC, a spreadsheet with ten columns (Appendix 3) was created. The first column contains the city’s name, the second, third, fourth, fifth, and sixth columns contain the city’s rank based on the cited reports, the seventh and eighth columns are for the average and standard deviation of each city’s rank, and the ninth and ten columns contain the number of times a city appears in all five ranks (NTR) and the final rank.

The criteria to select the 25 benchmark SC are (1) being in at least three rankings; (2) the average of all rankings selected from the lowest to the highest score. For each best city, additional research was done in the scientific databases mentioned in phase 1 and the government site to examine their SC plan, strategy, program, project, model, roadmap, main terms used to define a SC, city’s vision, and approaches to engaging citizens over time.

Literature, bibliometric, and documentary review

The review was organized by the following topics: Climate change as a global challenge, Brazil CO2 emissions, Brazilian Amazon region CO2 emissions, Brazilian Amazon State and Manaus Profile, human settlement, and urban cities, publications from cities in evolution to SC, P5 model with enablers that are contributing to the popularization of SC, SC definitions, profile of benchmark SC, DT, and digital technologies with approaches to CE.

Climate change as a global challenge and Brazil CO2 emissions

Globally, Rockström et al. (2009) showed that nine systems regulate the stability and resilience of our planet. They proposed a quantitative planetary boundary, under which humanity can develop itself for generations but crossing them might jeopardize life due to large-scale irreversible environmental changes. However, Steffen et al. (2015) and Persson et al. (2022), found that five of the nine planetary boundaries have already crossed due to anthropogenic activities, of which biosphere integrity and climate change are considered the main limits. Furthermore, a recent publication (Rockström et al., 2023) modified their original concept and showed that seven of eight thresholds—climate, natural ecosystem area, ecosystem functional integrity, surface water, groundwater, nitrogen, phosphorus, and aerosols—have been crossed.

In terms of climate change, it was learned that human emissions of CO2 and other greenhouse gases are its primary drivers (IPCC, 2013). In addition, from 1850 until 2021 (Table 1), it was estimated (Evans, 2021) that humanity has pumped around 2,500 bn tons of CO2 into the atmosphere with the USA (509 Gt), China (284 Gt), Russia (172 Gt), Brazil (113 Gt), and Indonesia (102 Gt) among the highest emitters.

Table 1 Nations with the largest cumulative CO2 emissions from 1850 to 2021.

Countries	Fossil & cement	Total land & forests	Total	
1st USA	420 Gt (82.5%)	89 Gt (17.5%)	509 Gt	
2nd China	242 Gt (85%)	43 Gt (15%)	285 Gt	
3rd Russia	117 Gt (68%)	55 Gt (32%)	172 Gt	
4th Brazil	16 Gt (14%	97 Gt (86%)	113 Gt	
5th Indonesia	15 Gt (15%)	88 Gt (85%)	103 Gt	
Total	810 Gt	372 Gt	1,182 Gt	
Note:

Source: Evans (2021).

Concerning Brazil, it was found (Evans, 2021) that the country represents 4.52% of the global CO2 emissions, with an interesting discovery related to the origin of CO2 emissions, divided into two groups, one related to emissions from fossil fuels (including cement) and another related to emissions generated from land and forests. Table 1 shows that Brazil’s main cumulative CO2 emissions are from land and forests (97 Gt; 86%), along with Indonesia (88 Gt; 85%), while for the USA (420 Gt; 82.5%), China (242 Gt; 85%), and Russia (117 Gt; 68%) it comes from fossils and cement.

The global carbon emissions have increased considerably in the last decades, and when Brazil’s CO2 emission is considered during the last three decades, from 1990 until 2020, it is estimated a total of 49.18 Gt of CO2, from which most (75.72%) is from land use and forest, while 19.12% from energy, 4.23% from industries, 0.87% from agriculture, and 0.05% from waste (SEEG Brasil, 2022).

Brazilian Amazon region CO2 emissions

Brazil has six biomes, Amazon Region Forest (420.8 Mha; 49.5%), Atlantic Forest (110.7 Mha; 13%), Cerrado (198.5 Mha; 23.3%), Caatinga (86.3 Mha; 10.1%), Pampa (19.4 Mha; 2.3%), and Pantanal (15.1 Mha; 1.8%).

The Brazilian Amazon Region is composed of nine states (Acre, Amazon, Amapá, Maranhão, Mato Grosso, Pará, Roraima, Rondônia e Tocantins) with a total area of 503,013,724 hectares, representing 59% of Brazil’s area (MapBiomas Brasil, 2022; IBGE, 2020). This region is a particular concern due to the possibility of reaching a tipping point that could exacerbate environmental problems (Ribeiro et al., 2022; Boulton, Lenton & Boers, 2022; Amigo, 2020; Nobre & Borma, 2009). A study carried out by MapBiomas Brasil (2022), revealed, that between 1985 and 2021:

(a) Brazil lost 84.7 Mha (millions of hectares) of native vegetation, mostly in the Amazon Region (44.1 Mha, ten times the size of RJ State), followed by the Cerrado (28 Mha), Caatinga (6 Mha), Atlantic Forest (1 Mha), Pantanal (0.7 Mha), and Pampas Biome (0.1 Mha).

(b) In the last 36 years, Brazil burned at least 167.3 Mha (20% of the country), an area larger than Iran, including 73.4 Mha in the Cerrado, 69 Mha in the Amazon Region, 8.8 Mha in the Caatinga, 8.6 Mha in the Pantanal, 7.1 Mha in the Atlântica Forest, and 0.2 Mha in the Pampa. The five most critical states were Mato Grosso (38.9 Mha), Pará (21.5 Mha), Tocantins (16.6 Mha), Maranhão (15.5 Mha), and Bahia (11.6 Mha).

(c) In 36 years, the Amazon Region lost 11.73% of its native vegetation cover, especially due to pasture and agriculture, with most of the loss occurring in Pará State.

(d) The mining area increased by 600% in Brazil between 1985 and 2020, with 300% occurring in Conservation Units.

According to Terra Brasilis (2022), a Brazilian geographic data platform, from 1988 until 2021, 33 years, around 47,027,500 hectares were deforested in the Amazon Region, which represents 9.35% of the total Amazon Region Area.

Brazilian Amazon State and Manaus profile

Brazil has 27 states and the Amazon is the largest (1.56M km2), located in the North Region (IBGE, 2021b), with 3.94 m people living in 62 cities, most (52.3%) in the capital Manaus.

Manaus was founded on 24th October 1669 and in 2022 celebrated 353 years. It is among the capitals with the highest population growth in Brazil. According to the 2010 Brazilian National Census (IBGE, 2011a), Manaus had the seventh highest population of the 27 capitals, with 1.8 m inhabitants, and its population growth (%) was among the five highest since 1872, with the percentage higher than the average and median of the 27 capitals in almost all Census (Table 2).

Table 2 Evolution of the Brazilian capitals’ population growth (%)—compared to 2010.

Capitals	2010
1872	2010
1890	2010
1900	2010
1920	2010
1940	2010
1950	2010
1960	2010
1970	2010
1980	2010
1991	2010
2000	
1. Palmas	.	.	.	.	.	.	.	.	6,844	841	67	
2. Boa Vista	.	.	.	.	.	1,548	986	667	308	99	42	
3. Macapá	.	.	.	.	.	1,834	749	354	183	122	41	
4. RBR	.	.	.	1,586	1,995	1,090	602	296	180	71	33	
5. Manaus	6,043	4,554	3,483	2,280	1,594	1,191	928	474	180	78	28	
6. PVH	.	.	.	.	.	1,473	739	382	210	50	28	
7. Brasília	.	.	.	.	.	.	1,713	371	114	61	26	
8. Aracajú	5,875	3,396	2,603	1,426	868	629	394	206	91	42	24	
Average →	5,519	3,975	3,284	1,569	1,029	840	521	244	357	74	20	
Median →	3,107	2,927	2,461	1,412	923	679	394	211	109	41	17	
–	–	–	–	–	–	–	–	–	–	–	–	
27. POA	3,103	2,589	1,813	686	418	258	120	56	22	12	4	
Note:

Source: Author based on IBGE (2011b).

In terms of Economy, the Amazon State’s activities are organized into four main sectors: (a) agricultural (livestock, forest production, fisheries, and aquaculture, etc.); (b) industry (extractive, transformation, construction, electricity, and gas, water, sewage, waste management activities, waste, and decontamination); (c) service (trade, transport, accommodation and food, information and communication, education, art, culture, etc.).

According to SEDECTI (2022a), Amazon’s GDP in 2021 was R$ 126.31 billion, a nominal growth of 16.93% from 2020. Industry (30.1%) and Services (48.8%) sectors grew by R$ 38 billion and R$ 61.5 billion, respectively.

Concerning jobs, the total employment contracts in effect on December 31st, 2021, in the Amazon State, reached the mark of 447,386, around 6.27% higher than recorded in 2020, most (44.6%) allocated in Service, followed by industry (25.6%), trade (23.8%), agriculture (8.4%), and construction (5.2%).

In addition, among the 62 Amazon State cities, most (408,972; 91.4%) formal jobs were concentrated in Manaus, followed by the cities Itacoatiara (4,822; 1.07%), Presidente Figueiredo (3,408; 0.76%), and Manacapuru (3,218; 0.71%) (SEDECTI, 2022b p. 8–9).

Manaus’ population growth and job concentration are due to the Amazon Rubber Boom (Bradford Burns, 1965; Resor, 1977) and later to its free import/export area with special fiscal incentives, a model called the Manaus Free Economic Zone, now known as Manaus Industrial Park, composed of 600 industrial companies that recorded an annual revenue growth of 28.84% between 2021 and 2020 (Fig. 2), increasing from US$ 22.8 bi in 2020 to US$ 29.4 bi in 2021 (Suframa, 2022b).

Figure 2 Manaus industry park annual revenue from 2017 until 2021.

The main Manaus Industry Park sectors, in terms of annual revenue growth (Fig. 3) in 2021, are IT goods, electro-electronics, two-wheel pole, chemistry, and thermoplastics, with a total of US$ 23 bi, representing 78.4% of all sectors annual revenue.

Figure 3 Manaus industry pole main sectors annual revenue in 2021.

The Manaus Industry Park’s five lines of products with the greatest prominence in 2021 are:

(1) LCD screen TV with 10,347,458 units; (2) cell phones (14,451,800); (3) motorcycle, motor net, and mopeds (1,215,775); (4) mounted printed circuit board for computer use (182,481,598); (5) split system air conditioner (5,883,771), of which the annual revenue is shown on Table 3.

Table 3 The five main products of Manaus Industry Park in 2021.

Products	Production (Units)	Revenue (US$)	
1st Screen LCD TV	10,347,458	4,273,228,503	
2nd Cell phone	14,451,800	2,849,794,676	
3rd Motorcycle, motornet and moped	1,215,775	2,805,603,992	
4th Mounted printed circuit board for computer use	182,481,598	2,290,256,395	
5th Split system air conditioner	5,883,771	1,616,259,255	
Note:

Source: Suframa (2022b p. 11).

The city has grown rapidly in the last 60 years, driven by Manaus Industry Park, but unplanned urban expansion has caused several issues, including low garbage collection and recycling, polluted streams (Informe Manaus, 2022), poor sanitation (Instituto Trata Brasil, 2022), poverty, criminal violence, and CO2 emissions, with climate change worsening inequality, as described below:

(a) From 2000 to 2019, Manaus emitted at least 115,378,658 tons of CO2, an increase of 127.43%, most (90.2%) from energy generation, followed by land and forest use (9.8%) and waste (0.04%). For Shrivastava et al. (2019) Manaus is the Amazon’s main anthropogenic aerosol source during the wet season.

(b) Climate change can create vulnerabilities and lead to increased precipitation, heat (Geirinhas et al., 2017), pandemic and epidemic risks (SBMT, 2021; Mourão et al., 2015), shortages, higher prices, poor air quality, and extreme weather events like storms, droughts, and river floods (G1 AM, 2021; Espinoza et al., 2022).

These events harm everyone, increasing vector-borne diseases, water pollution, and food instability. According to Filho et al. (2021), these vulnerabilities, mediated by racism, poverty, geographic and cultural contexts, differ by race and ethnicity, exacerbating gender inequalities. Indigenous and black people from Manaus have the lowest water availability. There is also an unequal proportion between genders, with worse indicators for women.

(c) Poverty: in 2019, 47.4% of the population lived in poverty in the Amazonas State, a percentage higher than the rest of the region and higher than the rest of the country (Amazônia Legal em Dados, 2020).

The Manaus Municipal Human Development Index is 0.737, the 23rd lowest of the 27 capitals (Atlas do Desenvolvimento Humano no Brasil, 2020), while having the sixth highest GDP in Brazil in 2019 (IBGE, 2021c, p. 3).

Despite its wealth, Manaus concentrates 20% of the Amazon state’s population living in extreme poverty (UFAM, 2019), with 360,596 (41.29%) households (from a total of 873,410), allocated in subnormal agglomerations (invasions, slums, stilt houses, inadequate housing constructions) (IBGE, 2022).

(d) High degree of violence: Manaus was the second most violent city in Brazil in 2021 with 1,060 homicides, up 55% (685), 26% (839), and 19% (892) from 2020, 2019, and 2018 (SENASP, 2022). Manaus was also one of the 50 most dangerous cities in 2022, according to Consejo Ciudadano para la Seguridad Pública y la Justicia Penal (2022).

Human settlement and terms developed to face urban challenges

Since time immemorial, human beings have had to act collectively to overcome the difficulties inherent to their survival, establishing different types of coexistence and settlements, evolving according to political, technological, climatic, and population changes.

Historically, since our ancestors came from caves and/or forests, we live in families, and as the settlement population grows, families can become clans, tribes, villages, towns, cities, and cities can be transformed into urban agglomerations in several different ways (UN, 2019a), depending on each country or regional policies.

Human settlement is a place where people live, assuming many forms, it can be permanent or temporary, rural, urban, mobile, or sedentary, disseminated, or agglomerated (Živković, 2019). According to OCDE (2001), it is an integrated concept that comprises: (a) physical components of infrastructure and shelter; and (b) services in which the physical elements provide support to the community such as culture, education, health, recreation, nutrition, and welfare.

In terms of urban cities, the last World Urbanization Prospects 2018 (UN, 2019b) revealed that:

(a) Globally, more people live in urban areas than in rural areas, with 55% of the planet’s population living there in 2018, and by 2050, this percentage could be up to 68%.

(b) By 2030, the world is projected to have 43 megacities (10 m people), most in developing regions.

(c) as the world continues to urbanize, sustainable development depends increasingly on the urban growth’s successful management.

(d) To ensure that the benefits are shared, policies to manage urban growth need to ensure access to infrastructure and social service for all.

As the urban population is increasing over time, at an unprecedented rate, generating many problems, it has received the attention of authors from different backgrounds concerned with urban city planning, such as the journalist, editor, and writer Charles Mulford Robinson (1869–1917), known as a leader in the city planning movement in the USA, and his ability and collections supported the City Beautiful Movement in the USA (Yalzadeh & Blumberg, 2019), and raised public interest in the early 1900s on topics related to visual aspects of cities, civic beauty, control of its utilities (overhead wires), care and planting trees, etc. (Shillaber, 1967).

Another author is the landscape architect and editor, Frederick Law Olmsted (1822–1903). He was known as the founder of American landscape architecture, an active author on city planning (Library of Congress, n.d), with a belief that everyone should be able to visit and enjoy parks (Clinton, 2022), and probably the first to mention the term “Intelligent City” when asking and answering the question “How are we to further the progress of Intelligent City Planning?” (De Forest et al., 1912, p. 370).

Another pioneering author is the biologist, sociologist, and town planner Patrick Geddes (1854–1932), which the classical book “Cities in Evolution” contributed to Urban Planning, Environment, and Citizenship, raising reflections (Geddes, 1915) on:

(R1) General urban trends in a period marked by ugly, unsanitary cities with a waste of resources.

(R2) City planning, it should be taken seriously with the active participation of residents.

(R3) History, for each city, there is a need for a systematic survey of its development and origins, its history and its present, which requires not merely information on material buildings, but also the city’s life and its institutions.

(R4) The importance of the effective use of an interdisciplinary scientific approach to identify and solve city problems, survey them individually, and compare it with others.

(R5) The importance to see a city as an organism, not as a mechanical system, demanding citizens understand their city’s history, as well as be protagonists through a regional and civic survey during the city planning process (Geddes & Stalley, 1972; Garau, Zamperlin & Balletto, 2016).

Many authors also provided contributions such as Doxiadis (1963), Davis (1965), Hunter (1966), Johnson (1979), Mulliner (1979), Oleg (1982), Lipman, Sugarman & Cushman (1986), Kodama (1987), Gilb, Tarr & Dupuy (1989), Gibson, Kozmetsky & Smilor (1992), Heng & Low (1993), Carter (2004), Musterd (2004), Husieva, Kucheriava & Suptelo (2017), Ohno (2008), Shin (2009), Bibri & Krogstie (2017), Peris-Ortiz, Bennett & Pérez-Bustamante Yábar (2017), Al-Turjman (2019), Janik, Ryszko & Szafraniec (2020), Shirowzhan & Zhang (2020), etc.

As a result, hundreds of scientific articles, books, book chapters, conference articles, etc. have been published since 1900, with several terms developed to face different urban challenges, such as: Ecumenopolis, Dynapolis, Digital or Virtual City, Global or World City, Intelligent City, Low Carbon, Netzero/Net0 City, Knowledge City, Networked City, SC, Eco, Green or Sustainable City, Technopolis, Ubiquitous or U-City, as shown in Table 4.

Table 4 Terms applied to find articles, news, books, book’s chapter, and conference.

Terms in title or abstract	Main queries used	
Ecumenopolis	“Ecumenopolis”	
Dynapolis	“Dynapolis”	
Digital or Virtual City	“Digital City” OR “Virtual City”	
Global City or World City	“Global City” OR “World City”	
Intelligent City	“Intelligent City”	
Low Carbon or Netzero or Net0 City	“Low Carbon City” OR “Netzero City” OR “Net0 City”	
Knowledge City	“Knowledge City”	
Networked City	“Networked City”	
Smart City	“Smart City” NOT “Smart Residence” NOT “South Broward” NOT “Street Smart!” NOT “Smart City Car” NOT “M U D on a Street Car” NOT “Smart city coupe” NOT “IVY Brand Smart City Ddsy Hes Medidor de Sistema para Francês”	
Eco or Green or Sustainable City	“Eco City” OR “Green City” OR “Sustainable City” NOT “Impounded-Storage Requirements” NOT “Halal Tourism”	
Technopolis	“Technopolis”	
Ubiquitous City or U-City	“Ubiquitous City” OR “U-City” NOT “University City” NOT “U City Public Company” NOT “u. City of New York” NOT “Gif u city” NOT “Cities of Culture”	

After a bibliometric study (Gomes da Silva, 2022b), realized between 19th July 2022 and 25th August 2022, in five scientific databases, using terms and queries listed in Table 4, it was found (Fig. 4) that, among 19 terms related to urban city issues published between 1900 and 2021, SC received the highest number of publications (Lens = 29,725; Dimensions = 28,973; Engineering Village = 10,406; Web of Science = 8,564; and Science Direct = 2,494).

Figure 4 No. of publications related to urban city terms in five scientific platforms (1900–2021).

The next terms are Eco, Green, or Sustainable City (Lens = 7,965; Dimensions = 6,567; Engineering Village = 1,069; Web of Science = 1,618; and Science Direct = 1,199), followed by Global or World City (Lens = 7,820; Dimensions = 6,070; Engineering Village = 314; Web of Science = 1,319; and Science Direct = 544), and Digital or Virtual City (Lens = 2,308; Dimensions = 1,580; Engineering Village = 956; Web of Science = 823; and Science Direct = 123).

These findings suggest that SC is a topic of significant interest in urban city research, which is not surprising given the increasing adoption of technology and data-driven approaches to urban management. In addition, the prominence of Eco, Green, Sustainable, Global or World City also suggests a growing focus on sustainable development and globalization in urban research.

Publications from cities in evolution to SC

The first consistent publications on Intelligent Cities or SC, as part of public and private investments to improve urban areas, took more than six decades since De Forest et al. (1912) discussed issues related to City Planning and Housing.

Additionally, Geddes (1915, p. 198), in his classical book, dedicated one chapter to comparing the German town planning and organizational approach against the style of Great Britain. He used the organization of the state railways as an example, emphasizing how the German system (information, reservations, maps, tables, rates, and structure) was organized in an intelligent way that facilitated public cooperation and the economy of time and labor.

According to a literature analysis, the term Intelligent City has been used alongside others since the 1980s, whereas SC has garnered popularity in the last 22 years. The initial publications describe US, Japanese, and HK experiences, with emphasis on Singapore, in the following order:

Bloom & Asano (1981, p. 3, 5, 15) and Edginton (1989) examined formal Japanese Government National Policies, Strategies, Programs, and endeavors to revitalize the national economy through urban programs that emphasize innovation and technological development infrastructure. Examples include Tsukuba Science City (since 1963) and Kansai Science City (since 1978) to improve national public R&D, followed by the Technopolis Program (early 80s) to increase research and high-tech production in rural areas, Teletopia (since 1985) to introduce new computer-based information system in rural towns, and the Intelligent City (since 1986) to promote optic fiber and intelligent building principles in urban redevelopment, etc. Although Edginton (1989) noted several program challenges and a lack of cooperation between national and local governments, it was considered that a mechanism exists to support the Japanese Fourth National Comprehensive Development Plan.

Lipman, Sugarman & Cushman (1986) wrote the book “Teleports and the Intelligent City” about Teleports, satellite telecommunication center that switch data, video, and voice, utilizing steerable and frequency-agile satellite antennas. To the authors, Teleports could serve as a hub of the Intelligent City, combining information and communication technologies (ICT) with other elements to create a multidimensional “Intelligent” real estate community that can access satellites and provide solutions like Smart Buildings, Smart Parks, teleconferencing, facsimile transmission, etc.

It was an American real estate economic development investment, implemented in the early 80s, along with a distribution of fiber optic networks, with projects such as Atlanta Teleport (Douglasville), The Kansas City Teleport (Kansas), Staten Island Teleport (NY Port), The Bay Area Teleport (Harbor Bay Isle Business Park/California), Central Florida Teleport (Ocala Airport Commerce Center/Florida), Pacific International Teleport (Los Angeles) etc.

Kodama (1987) published an article in Japanese, titled in English “Information Systems in the aging society—some problems in Intelligent City”. A new expression, “Intelligent building” or “Intelligent City” in Japanese was proposed concerning the hardware side, and the author rethought the impact of IT on everyday living, especially for elders.

Concerned with the sophisticated information society in Japan, Akihiko & Osamu (1990) addressed intelligent building systems recommended by Japanese NTT and the future of those building systems in the Integrated Service Digital Network age.

Batty (1990) in an editorial article, explored the rise of information networks in urban cities, the network or informational city, the different approaches to developing national telecommunications, and IT policies in two countries (Singapore and HK), and the need for further discussion about the development of the intelligent city through information infrastructures.

Cheung (1991) focused on the relationship between the Japanese government’s promotion of regional information development policies (Teletopia, New media community, Greentopia, Intelligent city, etc), and the Fourth Comprehensive National Development Plan (1989–2000), with possible impacts on the National Land System.

Gibson, Kozmetsky & Smilor (1992) were probably the first to publish a book “The Technopolis Phenomenon—Smart Cities, Fast System Global Network” which mentions on the term “Smart Cities” with 216 pages focusing on Technopolis, a term widely used in Japan since the early 1980s (Yazawa, 1990), when the Ministry of International Trade and Industry (MITI) started to formulate a 10-year vision to develop industries and regions by using Technopolis as a strategy. The authors saw technology as a tool for political, economic, and social change. Although the book is not about SC, it discusses global smart infrastructure, smart office buildings in the US, Japan, Germany, and the UK, technology breakthroughs and human resources to accelerate high-technology development, information technologies, telecommunications, computer-based networks, and the Internet, which are the foundation of SC Development.

Jussawalla, Heng & Low (1992) discuss actions that are making Singapore an Intelligent City-State, such as telecommunications investments, the rise of the IT sector, which has created an infrastructure for global services, and the impact of multinational corporations on the new division of labor, which formed human resources skills. Heng & Low (1993) argue that government strategies and companies made Singapore an Intelligent City.

To Julian (1995), SC involves several types of projects to create high-tech islands by concentrating communication resources within a region, city, or district, providing the development of applications and communication technologies. Because it operates within present technologies, SC can serve as a model for the future communications environment.

Tan (1998), Mahizhnan (1999), and the Singapore Government (2015) address how Singapore is becoming a SC, through policies, plans, programs, projects, investments in people, infrastructure, and smart technologies that have been implemented on the island since 1963.

P5 model with enablers that are contributing to the popularization of SC

A bibliometric study using the queries shown in Table 4 in five scientific platforms to identify the total number of publications (journal articles, books, book chapters, and conference articles) from 1992 to 2021 found (Fig. 5) that the term SC has gained popularity in the academic community since 2009, increasing from 12 publications in 2009 to 6,222 publications in 2021, only considering publications on the platform Lens.org.

Figure 5 Facts and number of publications related to Smart City in five scientific platforms.

The SC term has benefited from investments developed by other urban city experiences, especially in ICT infrastructure, products, and services, and it gained continuous popularity not only in the academy, but also among police makers, companies, and investors due to enablers explained ahead.

In addition, some facts also listed in Fig. 5 and Appendix 4 are part of enablers that facilitate the spreading of SC initiatives around the world, with examples taken from the most active regions such as the USA, EU, Japan, China, South Korea, and Singapore.

In short, leadership, long-term vision, national and/or local policies, strategies, programs, projects, budgets, funds to support SC, improvement of ICT infrastructure, technological innovations (4G/5G, Smart phones, Cloud, etc), dissemination of best practices, national and international events, innovation ecosystem, technology, alliances, transparency, and business result such as increased access to the Internet, devices, and IT services, are among the enablers that are contributing for spreading SC initiatives around the world.

As an example, Fig. 5 shows that SC publications increased after Telia Company (2009) launched 4G in Sweden and Norway in 2009 and IBM launched the Smarter City Campaign in the US. The National U-City Plan 1 in South Korea, with an R&D fund of 1,017 billion won (752 billion from the government and 265 billion from the private sector) was launched shortly after.

However, when authors discuss SC enablers, most focus on technical issues related to ICT (Puron-Cid & Gil-Garcia, 2022; Czupich, 2019; Lučić, Weber & Lovrek, 2016; El et al., 2016), or technologies like IoT (Shah & Mishra, 2016; Santos et al., 2018; Evertzen, Effing & Constantinides, 2019; Peneti et al., 2021a, 2021b), 4G or 5G (Lynggaard & Skouby, 2015; Loghin et al., 2020), Big Data (Vuppalapati et al., 2017; Bergamini et al., 2018), Cloud (Tei & Gurgen, 2014), AI (Nikitas et al., 2020; Soomro et al., 2018; Bhushan et al., 2022), Blockchain (Hakiri & Gokhale, 2021; Kim et al., 2022), and so on.

While ICT infrastructure and technologies are relevant to facilitate the implementation of SC initiatives, a strategic perspective that places citizens at the center must be considered. A more holistic classification could be composed by a P5 model, as shown in Fig. 6, in which enablers are managed by the following stakeholders: public organizations, protagonist people (CE KIT), and private organizations & partners.

Figure 6 P5 model with enablers that are contributing to the popularization of Smart Cities.

The P5 model was created based on the review of the aforementioned articles, and based on:

(a) Other articles (Boniotti, 2021; Jiang, Geertman & Witte, 2022; Fernandez-Anez, Fernández-Güell & Giffinger, 2018; Chourabi et al., 2012; Kogan & Lee, 2014; Fernandez-Anez, 2016; Ferrer, 2017; Azevedo Guedes et al., 2018; Mahizhnan, 1999; Kristiningrum & Kusumo, 2021).

(b) Official portals and documents from Governments (BIS, 2013; China Daily, 2013; Ville de Montréal, 2015; Singapore Government, 2015; Taiwan Ministry of Digital Affairs, n.d.; South Korea Government Ministry of Land, Infrastructure and Transport, 2020; UK-Asean Business Council, 2021 p. 159; Japan Cabinet Office, MIC, MLIT and Smart City Public-Private Partnership Platform Secretariat, 2021; Seoul Metropolitan Government, 2022; Van Coung, 2022).

(c) other types of organizations (IESE Business School, 2022; Open North, 2018; ITU, 2019; Yukiko, 2021; Eden Strategy Institute, 2021; IMD SC Observatory and SUTD, 2022; Fira Barcelona, 2022; WBG, 2018) that deals with SC issues.

Figure 6’s content is not exhaustive nor are the enablers exclusive to each stakeholder. It's basically a way to properly classify the enablers that are helping SC initiatives gain popularity.

Enablers are drivers that contribute to generating a shared vision, trust, motivation, teamwork, collective participation, fact-based decisions, standards, correct attitudes, prizes (awards), alliances, technical and financial support, scientific approaches, and multidisciplinary solutions to address the city’s most pressing problems and challenges.

For example, the public organizations enablers are leadership, development of a long-term vision and principles to inspire all city stakeholders, as well as diagnostics, priority areas, goals, policies, programs, roadmaps, projects, budgets, incentives, ICT infrastructure, transparency, legal frameworks, innovation ecosystem, education & training, alliances, etc.

According to the Eden Strategy Institute (2021), based on the public organization enablers mentioned in Fig. 5 (they refer as factors), an extensive study involving 235 cities around the world revealed that the top ten SC Governments for 2020/2021 are Singapore, Seoul, London, Barcelona, Helsinki, New York, Montreal, Shanghai, Vietnam, and Amsterdam.

In the other ranking, when the perceptions of the population of 118 cities are taken into consideration on issues related to technology applications and infrastructure available to them, the ten best SC are Singapore, Zurich, Oslo, Taipei, Lausanne, Helsinki, Copenhagen, Geneva, Auckland, and Bilbao (IMD SC Observatory and SUTD, 2022).

When the enablers “Best Practices, Annual Events, and Awards” are considered, the SC Expo Congress is a good reference. To have an idea of the event’s impact, in 2022, it attracted 28,621 online attendees, 20,402 in-person attendees, 853 exhibitors, and more than 400 speakers from 134 countries. Furthermore, for the World SC Award referees, the best SC in 2022 are Seoul, Kyiv, Bogota, Curitiba, Sydney, and Toronto (Fira Barcelona, 2022).

The recognition of the best countries and/or cities is the result of a long-term investment, involving collaboration among public, private, and NPOs, as shown in Fig. 5 and Appendix 4. At the center of the P5 model is the protagonist people to make citizens active actors in the management of a SC, by using the CE KIT, whose model, methodologies, and approaches to engage citizens are explained in “Proposed Participatory SC Model (CE KIT)”.

SC definitions

A bibliometric study using the Lens.org platform and the query “Smart City Definition” to find publications (books, book chapters, conference proceeding articles, journal articles, and conference proceedings) with titles and abstracts containing this phrase, revealed 63 publications with the following profile (Gomes da Silva, 2022c):

Publication types: most articles (29; 46%) are published in journals, followed by 18 book chapters (28.6%), 12 conference proceeding articles (19%), and four books (6%).

Main authors: Renata Paola Dameri has the most publications, with one article and three book chapters, followed by Felipe Moura and Joo de Abreu e Silva, each with three book chapters.

Highest citations: Albino, Berardi & Dangelico (2015) received 1,816 citations, Dameri (2013) received 339 citations, Lai et al. (2020) received 94 citations, and Nikitas et al. (2020) received 86 citations.

Fields of study: the ten main fields classified by the Lens.org platform include SC (54), followed by business (22), sustainability (17), computer science (15), corporate governance (12), architectural engineering (12), urban planning (11), governance (10), and engineering (10). Notably, fields such as citizen engagement (2), open data (2), information technology (2), transparency (2), quality of life (2), circular economy (1), big data (1), creativity (1), are among the least mentioned.

Among the publications, the following considerations are notable:

Dameri (2013) claims that SC is a bottom-up phenomena and that citizens should be the most essential topics in its definition, yet they are often ignored. The author also defined a SC after a literature analysis and considering four key aspects: terminology, components, boundaries, and scope: A well-defined geographical area, in which high technologies such as ICT, logistics, energy production, and so on, cooperate to create benefits for citizens in terms of well-being, inclusion and participation, environmental quality, and intelligent development; it is governed by a well-defined pool of subjects, able to state the rules and policy for the city government and development.

Russo, Rindone & Panuccio (2014) presented the evolution of five definitions and argued that a SC should include people to ensure residents and stakeholders participation. They also noted that a top-down strategy encourages cooperation, whereas a bottom-up one allows more direct participation.

To learn about theoretical and practical cases involving top-down and/or down-up approaches, it is recommended to read Capra (2014), Ville de Montréal (2015) and Leu et al. (2021) with successful cases reported in Amsterdam, Montreal, and Taipei.

Albino, Berardi & Dangelico (2015) also presented the evolution of the SC definition, analyzing 23 definitions, and argued that the SC concept is no longer limited to the diffusion of ICT, but it looks at people and community needs.

Although the authors did not propose a definition, they pointed out that the term is missing people, and they believe that people should be the protagonist of a SC, shaping it with continuous interaction. They reflected on creativity, education, training, learning, culture/arts, and viewed SC as magnets for creative people, creating a virtuous circle making them smarter.

Fernandez-Anez (2016) and the International Telecommunication Union (ITU, 2014) established two methodologies to define a comprehensive SC and Smart Sustainable City, respectively. Both can help people build an interdisciplinary and scientific approach to learn from others and develop a better definition for their city, as mentioned below.

In the first case, Fernandez-Anez (2016) used a methodology with three phases, followed by a literature review, text analysis tagging technics, and descriptive statistics to identify 32 different SC definitions, as well as 404 terms, classifying them by:

(a) Four stakeholder types (seven universities, eight companies, five governmental institutions, and eight local governments working with SC) by using the knowledge-based helix model.

(b) Six SC characteristics developed by Giffinger et al. (2007) and European Smart Cities (2008): Smart Economy, Smart Environment, Smart Governance, Smart Living, Smart Mobility, and Smart People.

(c) SC main goals: efficiency, sustainability, and quality of life (QoL).

(d) Technological approach composed of ICT, connection, technology, tool, information.

(e) Others composed by city, data, innovation, equity, stakeholders, etc.

The author’s analysis revealed differences in how stakeholders define “Smart City” in their work:

Academia emphasized people, governance, ICT, connection, and environment.

Government institutions focused on governance, environment, people, ICT, and sustainability.

Local government highlighted people, governance, economy, environment, technology, and innovation.

Private organizations used a balanced approach covering connection, city, governance, efficiency, environment, living, economy, innovation, technology, sustainability, people, and QoL.

Across all stakeholders, people and governance were most mentioned, followed by environment.

As a result, Fernandez-Anez (2016) proposed the following definition of a SC: A system that enhances human and social capital wisely using and interacting with natural and economic resources via technology-based solutions and innovation to address public issues and efficiently achieve sustainable development and high quality of life based on a multi-stakeholder, municipally based partnership.

Another conclusion focuses on adopting a citizen-centric approach. Specifically, the author recommends increasing awareness and participation among civil society and individual citizens. Opportunities should be created for residents to share their perspectives and visions, which can then be incorporated into SC development.

Now the second case is presented: an analysis developed by a partner of The United for Smart Sustainable Cities (U4SSC), the International Telecommunication Union (ITU, 2014). The analysis was based on 116 definitions of Smart Sustainable Cities found in different sources. Using keyword analysis and grouping, the ITU identified 30 key terms to be included in the standard. These key terms were classified into eight key groups and six categories, as shown in Table 5.

Table 5 Results of the ITU (2014) smart sustainable definitions analysis.

Key groups	Key categories based on KI	
G1: ICT, Communication, Intelligence, Information	C1: Smart living	
G2: Infrastructure and services	C2: Smart people	
G3: Environment, Sustainable	C3: Smart environment, Sustainability	
G4: People, Citizens, Society	C4: Smart governance	
G5: Quality of life, Lifestyle	C5: Smart mobility	
G6: Governance, Management, Administration	C6: Smart economy	
G7: Economy, Resource		
G8: Mobility		

At the final, they proposed the following Smart and Sustainable City definition: An innovative city that uses information and communication technologies (ICTs) and other means to improve quality of life, the efficiency of urban operations and services, and competitiveness, while ensuring that it meets the needs of present and future generations concerning economic, social, and environmental aspects.

In addition, it is important to note that a definition is a statement expressing the essential nature of something (Merriam-Webster, 2022) and that a good definition should be simple and clear to concisely explain something. Taking this into consideration, a good example was developed by Smart Cities Council (2015), as part of a complete planning manual that is helping practitioners create a SC Vision and action plan. For them a SC is: A city that uses ICT to enhance its livability, workability, and sustainability.

According to the Smart Cities Council (2015), a SC first acquires data about itself using sensors, devices, and systems. Data is sent over wired or wireless networks. The data is then analyzed to identify present and future events.

This is an interesting definition that is straightforward, concise, and well-explained.

Based on the above, it is recommended that before starting on any SC journey, decision makers should study several definitions and select the most suitable for their reality. And the selected definition should then be used to guide the development of the desired type of SC (vision), followed by strategy, program, master plan, roadmap, framework, projects, budget etc. For this reason, “Profile of Benchmark SC” focuses on the definitions and visions developed by the best SC worldwide.

Profile of benchmark SC

Based on the five specialized reports that evaluated more than 200 cities worldwide and the criteria explained in Phase 4 of the Survey Methodology, it was possible to identify valuable information from the best 25 Benchmark SC listed in Appendix 5, as well as develop the CE KIT Model described in “Proposed Participatory SC Model (CE KIT)”.

Their profile is described as follows:

(First) Benchmark SC.

The twenty-five best SC are (1st) Amsterdam (X = 7.6; S = 6.5), (2nd) Singapore (X = 8.6; S = 10.26), (3rd) New York (X = 8.8; 6.26), (4th) London (X = 8.8; S = 8.93), (5th) Helsinki (X = 10.5; S = 6.86), (6th) Seoul (X = 10.8; S = 8.35), (7th) Copenhagen (X = 12.2; S = 13.18), (8th) Oslo (X = 13.0; S = 12.49), (9th) Vienna (X = 16.5; S = 9.98), (10th) Washington (X = 16.67; S = 15.95), (11th) Zurich (X = 18.0; S = 18.71); (12th) Berlin (X = 19.4; S = 18.96), (13th) Sydney (X = 21.4; S = 11.04), (14th) Taipei (X = 21.5; S = 13.23), (15th) Barcelona (X = 21.6; S = 23.35), (16th) Toronto (X = 22.67; S = 12.58), (17th) Paris (X = 23.5; S = 25.96), (18th) Madrid (X = 24.0; S = 8.12), (19th) Busan (X = 24.33; S = 16.26), (20th) Dublin (X = 25.5; S = 16.36), (21st) Melbourne (X = 25.7; S = 10.69), (22nd) Los Angeles (X = 27.7; S = 10.86), (23rd) San Francisco (X = 27.5; S = 21.92), (24th) Hong Kong (X = 28.0; S = 12.12), (25th) Montreal (X = 30.0; S = 20.22).

(Second) Continental highlights.

In terms of continent, although SC are a global phenomenon with several regions exploring new solutions to tackle urban challenges, most benchmark SC are in Europe (12; 48%), a region that is considered a leader in SC investment and development, followed by North America (6; 24%), Asia (5; 20%) and Oceania (2; 8%).

(Third) Adoption of formal documents to manage SC initiatives.

Most (84%) cities have developed a formal document such as a program, master plan, strategy, blueprint, program, project, initiative, or project, while in only 16% of the cases (Copenhagen, Oslo, Toronto, and Melbourne), it was not possible to identify such documents.

These documents are essential for the effective governance and administration of the SC initiatives, and in a desirable format, they should include the definition of a SC, the vision, goals, means, projects, etc.

The pioneer cities in implementing formal documents include Amsterdam (Amsterdam SC Program, since 2009), Seoul (Smart Seoul 2015, since 2011), Vienna (The Big SC Wien Initiative, since 2011), Barcelona (SC Strategy, since 2011), Helsinki (Helsinki Smart Region, probably from 2012), London (Smarter London Plan, since 2013), Singapore (Singapore Smart Nation, since 2014), and Montreal (Montreal Smart and Digital City Strategy, since 2014). The early adoption of SC technologies and approaches to engage citizens by those pioneer cities has had a significant impact on the development of the SC movement around the globe, contributing to paving the way for a more efficient and sustainable urban life.

In contrast, the latest cities are Los Angeles (SmartLA 2028, since 2020), Sydney (SC Strategy Framework, since 2020), HK (HK SC Blueprint 1.0, since 2018), and Zurich (Strategic SC Zurich, since 2018).

(Fourth) Who led the SC initiative in the region.

It was found that in most cases (19; 76%), the initial leadership and investment for SC initiatives came from the local government partnering with the private sector, or vice versa. This highlights that collaboration between city officials and private companies is critical for implementing SC projects. Such partnerships provide important benefits like expertise, resources, accelerated innovation and investment, shared risks, and meeting the needs of inhabitants and businesses.

The Amsterdam SC Program pioneered this model in 2009 through a partnership between the Municipality of Amsterdam, grid operator Liander, and the Amsterdam Innovation Motor (AIM). These key stakeholders collaborated to launch projects focused on energy efficiency (Capra, 2014, p. 40).

It is worth mentioning the importance of organized civic society, since Marleen Stikker founded the Digital City on 15th January of 1994, the first virtual community with free public access to the internet in Amsterdam. The foundation of Waag Future Lab was important to reinforce the critical reflection on technology and encourage social innovation in the city.

In terms of private contributions, IBM has excelled among private companies that support SC initiatives. Its SC Challenge has supported districts and local governments to join or reinforce the SC journey over time. For instance, IBM supported the Helsinki Region Infoshare Program in 2010, the Juron Lake District in Singapore in 2011, Copenhagen and Taipei in 2013, Dublin in 2014, the Madrid Intelligence Project (MiNT) in 2015, Amsterdam and Melbourne in 2015, and Busan in 2017.

Furthermore, cities influenced by National Government leadership represent 20% of the cases such as Singapore, Washington, Taipei, Toronto, and San Francisco.

Only in Melbourne and Oslo, it was not found a formal plan, strategy, or roadmap dedicated to implementing a SC, as part of the City Hall initiative. However, this site <https://nscn.eu/Oslo> informs that Oslo City is implementing a wide range of SC projects, but it does not show any documentation to support it.

Concerning the City of Melbourne (2015), the city won the IBM SC Challenge, and in 2021, they launched the Economic Development Strategy 2031, and one of the key priorities for the city growth is Digitally Connected City (City of Melbourne, 2021a p. 30–31), with three actions related to investment in digital infrastructure, open data platform, libraries, etc. In the same year, the Community Engagement Policy and Melbourne Neighborhoods Planning Framework were approved, and they are unique documents to stimulate citizens’ participation.

(Fifth) Declaration of SC’s Definition.

Eighteen cities (72%) have declared at least one definition of SC in their documents or digital platforms, five (20%) did not, while two (Seoul and Busan) likely adopt definitions from South Korea’s advanced national policies and platforms that support SC.

The findings reinforce that a SC definition should be clear, precise, relevant, and publicly accessible. It would be counterproductive for city leaders tasked with SC implementation to proceed without first aligning on the meaning through stakeholder debates. Discussing and proclaiming an official definition helps stakeholders to develop and better comprehend the Vision, provide support, set expectations, and educate the public.

(Sixth) Declaration of SC’s Vision.

The vision answers the question “What kind of city do we aspire to be in the future?”, and local managers should design and declare a sound vision for the public because it shows the desirable city, sets the direction, sets goals for the city’s development, and provides a roadmap for the future.

A sound vision may motivate and mobilize the stakeholders (including citizens), and focus efforts and resources over time. Declaring a sound vision towards a SC contributes to attracting investment and partnerships from the private and other sectors.

When the 25-benchmark SC’s visions were investigated (Appendix 5), 23 (92%) have declared it at least once, which may vary based on the program’s deadline or the city’s new leadership. Using almost the same procedure as Fernandez-Anez (2016), various desired cities, goals, and means were found (Fig. 7).

Figure 7 Main terms used by the 23 benchmark cities in their vision.

Regarding desired city type, the most used terms were Connected (13 mentions), Digital (12), Smart (10), and Sustainable (5), followed by Open (5), Platform (4), Urban or Living Lab (4).

One reason cities like London, Oslo, Washington, and Zurich, aspire to become Connected Cities is that cities are complex, interconnected systems rather than isolated entities.

One reason cities like Singapore, NY, Seoul, LA, and Montreal envision themselves as Digital Cities or Platforms for digital transformation is the rapid advancement of technologies like IoT, AI, 5G, big data, and digital twins. These technologies allow cities to collect and analyze huge amounts of real-time data to inform decision-making and improve city management.

One reason cities like Berlin, Paris, Madrid, Helsinki, and Copenhagen envision becoming Sustainable or Carbon Neutral Cities is to address environmental challenges sustainably and enhance resilience, working toward decarbonizing their economies.

Some of the reasons cities such as Amsterdam, Oslo, or Dublin seek to become Open Cities include improving accountability and transparency, encouraging participation and collaboration, and fostering innovation and entrepreneurship.

Concerning goals to reach the vision, the terms cited have relation to QoL (24), sustainability (22), and efficiency (21). When the five fields are considered as goals to reach the vision, the terms most cited are related to people (24), economy (23), and environment (22), while mobility (3) is less mentioned.

Analysis of the means mentioned in the Vision statements revealed the most frequently cited terms were related to technology (20), connection (14), innovation (14), data (12), stakeholders (10), and ICT (7). They have contributed to data collection and analysis, connectivity, communication, energy efficiency, sustainability, intelligent transportation systems, public safety and security, CE, and more to make cities smarter.

Returning to the vision formulation, a good vision definition should be clear and inspirational, with a maximum of 30 words to be easily memorized and attract attention. While each benchmark city has its own approach to crafting its vision, an ideal statement should specify certain key elements. As proposed in Fig. 7, the most effective vision declares a deadline or timeframe, the desired Smart City type, goal(s), and mean(s) to achieve the vision.

In addition, it is highly recommended that the Smart City Definition and vision should be declared in an accessible public document such as act, law, strategy, policy, program, project, roadmap, blueprint, guidance, handbook, or official electronic platform (Website).

DT

As mentioned in “P5 Model with Enablers that are Contributing to the Popularization of SC”, several authors have written about ICT and technologies as the main enablers of SC, but they should be seen as a means not an end. Various technologies serve as means for achieving SC’s goals, selected based on specific initiatives and local needs. For example, in late 2017 the Government of Canada organized an SC Challenge, calling communities nationwide to develop bold solutions that would improve citizens' lives through data and connected technology.

This challenge garnered 225 total applicants, with 130 becoming eligible.

On June 1st, 2018, twenty were chosen as finalists, and according to the challenge organizers:

(a) The top ten technologies proposed were mobile applications (119), open data platforms (103), IoT (101), big data analytics (100), networks (99), geospatial (96), cloud computing (94), sensors (92), ai (83), enterprise solutions (73), and environmental monitoring (72).

(b) The five main areas were Empowerment and Inclusion (31%), Economic Opportunity (23%), Environmental Quality (13%), Healthy Living and Recreation (13%), and Mobility (12%).

Another example is Radu (2020) literature assessment on four DT—AI, big data, blockchain, and IoT—and their effects on SC’s components (economy, environment, governance, living, mobility, people). The author defined DT as “innovative solutions that require fewer resources and can grow exponentially, very often, shaking up the economy and structure of the related business”.

The results showed that: (1) DT in SC focus on mobility and transit, environmental sustainability, health, security, business efficiency, energy efficiency, and education; (2) AI, big data, blockchain, and IoT can improve SC if utilized responsibly; (3) AI, big data, and IoT automate decision-making and problem-solving and help construct smarter cities; (4) blockchain improves data security, communication, and legacy infrastructure and resource use; (5) DT can make cities smarter if individuals know and care about public and personal values (Radu, 2020, p. 1032–1034).

Digital technologies with approaches to engage citizen

The Government of Canada (2018) and Radu’s (2020) experiences demonstrate the growing relevance of data, internet, and DT in establishing digital platforms to solve city problems. Digital platforms also facilitate citizen participation through co-creation, feedback, surveys, voting, transparency in the city’s budget and project progress, efficiency, and the creation of innovative urban solutions.

As shown in Appendix 5, the 25 benchmark SC developed at least three digital platforms to engage citizens over time, focusing on living labs, mobile apps, official websites, and open data platforms, as summarized in the following sections.

Living labs

The origin of living lab is not from SC initiatives, but from universities, challenging students to learn courses by undertaking real-world projects in a community, dialoguing with several stakeholders, or putting themselves in the place of the customers, during the process of solving problems of the public.

Bajgier et al. (1991) created a course at Drexel University for students to develop and apply community Operations Research techniques by using a city neighborhood as a living laboratory. Their conceptual model of a living laboratory classroom setting was applied in a relevant commercial and residential area located in Philadelphia, providing students with a unique opportunity to participate in public policy projects.

Fisher (1995) developed the project “The Glanny Flat” at Kean College to introduce the principles of Universal Design to students and challenged them to develop an independent living environment for senior-age customers. Before the development of the design process, students visited the Oklahoma State University’s Barlett Independent Living Lab to each spend 5 min in a wheelchair and maneuver themselves in a space that simulates a common ranch-style residence with adaptive and/or assistive features involved in a universal design perspective.

McNeese (1996) from Pennsylvania State University used the living lab to integrate technology, context, and humans into a cyclical design process, while Bajgier et al. (1991) and Fisher (1995) used Operations Research and Universal Design with other disciplines to challenge students to solve real-life problems. In McNeese, Perusich & Rentsch (2000) and McNeese et al. (2005a, 2005b), the living lab framework integrates theory and practice to enable tool and technology development as a continuous process, with four components and practical cases:

(Component 1) Ethnographic studies (involving fieldwork, which may include living with the community, being studied, conducting interviews, observing daily activities, interactions, etc.).

(Component 2) Knowledge elicitation (a process of extracting and capturing knowledge and information from human experts or other sources).

(Component 3) Scaled worlds (virtual environments or simulated worlds that are designed to replicate the real world or other fictional worlds, but on a smaller or larger scale).

(Component 4) Reconfigurable prototypes (physical or digital prototypes that can be easily modified or reconfigured to test different design variations and explore different design options).

Other authors mentioned in the literature are:

Markopoulos & Rauterberg (2000) issued a white paper on Eindhoven University of Technology (TU/e)’s living lab, a platform for collaborative research initiatives to create and test home-related technology. They explored the living lab as a vacation on a campus, a temporary home, where “residents” are invited to experiment with novel technologies for 1 or 2 weeks, allowing TU/e’s research to investigate technology use in a situation near real life, reducing costs and providing observations that would be difficult to get in other situations.

A total of 5 years later, Intille et al. (2005) published another similar experience developed at MIT, a research facility called Placelab, located in a condominium building within Cambridge, MA neighborhood, a living lab considered as another tool for technologists, ethnographers, and others interested in studying and developing technologies that respond to home behaviors.

Living labs are one of six types of test and experimentation platforms (TEP) used by Ballon, Pierson & Delaere (2005) to test technology in real-world contexts and include end users as co-producers of ICT. Living labs like Kenniswijk (NL), Arabianranta Helsinki Virtual Village (FIN), and @PPLe (UK) present potential users with technology prototypes or demonstrators early in the innovation process, the authors say.

Living labs gained momentum when in November 2006, the European Network of Living Labs (EnoLL) was founded. Nowadays, it is composed of 155 active members from several countries with more than 480 recognized living labs.

For EnoLL, living labs are real-life test and experimentation environments that foster co-creation and open innovation among the main actors of the Quadruple Helix Model (academy, citizens, government, and industry). Its digital platform (https://enoll.org/) is a good source for those interested to develop a citizen-driven network empowering everyone to innovate with living labs projects developed for SC, education, design, creative industries, climate, etc.

Living labs also became popular when they supported SC initiatives, and the literature study showed that the first authors to write about that were:

(a) Dupont, Guidat & Morel (2010) mentioned the NIT SC Living Lab with two decades of piloting experience on complex projects to tackle societal issues (Nancy area) to establish a “user-driven” approach with residents and other actors’ participation to enhance citizen quality of life and support local economic development, contributing to the smart process in Smart Cities to make cities smarter.

(b) Schaffers et al. (2011a) advocated integrating living labs, future internet, and IoT platforms to create a Smart Cities experimental environment for service innovation. Three FP7-ICT project cases—ELLIOT (Experimental Living Lab for the IoT), SmartSantander (IoT experimental facilities in Santander city with over 20,000 sensors), and TEFIS (Future Internet Experiments)—were used. They argued that living labs can provide action research, development, data collection, user-driven application development, and user interaction to build collaborative partnerships for SC. Schaffers et al. (2011b) also discuss how living labs are helping SC evolve.

(c) Paskaleva (2011) examines the role of a SC as a link for open innovation by critically reviewing EU programs and SC Projects. The author focuses on EPIC, PERIPHRIA, and SMARTiP, where open innovation was developed as a key driver of the SC by considering the living lab ecosystem. The author believes that until 2011, living labs can provide the natural ecosystem for open innovation, but the traditional model, where civil servants act like private employees and engage with citizens as end-users, provides input for predetermined concepts or service models rather than proactively involving them in shaping the initial policy direction that determines service priorities.

SC and international networks have expanded numerous types of living labs worldwide. In six Finnish cities, Leminen, Rajahonka & Westerlund (2017) classified living labs and proposed a typology of the third generation, characterized by diverse platforms and participation approaches, resulting in four distinct models of collaborative innovation networks where the city could be viewed as a catalyst, neighborhood participator, provider, or rapid experimenter.

Analysis of the 25 benchmark SC revealed approximately 100 Living Lab initiatives, especially concentrated in the following cities, primarily located in Europe: (1) Amsterdam with seven Living Labs; (2) Barcelona (7); (3) Copenhagen (7); (4) Paris (6); (5) Berlin (6); (6) Singapore (5) and (7) Helsinki (5).

Mobile apps

They provide information and services, and engage citizens in a variety of ways, such as:

(Case 1) M-voting App, launched by the Seoul Metropolitan Government in 2017, to replace costly surveys, offline meetings, and town hall meetings to assess the feeling of inhabitants. It has been used to involve citizens not only to vote on the policy decision-making process but on any ordinary city life issues, by using a smartphone or a personal computer.

(Case 2) SmartAppCity, a global solution to SC, developed by Get-App in Spain, to help cities bring all information and services together, making them available to residents, city councils, shops, and businesses. It received several awards, for example, in 2013, it was the best app of the year at La Rioja Internet Award, and in 2014 received recognition during the Cities Summit London, and Madrid Smart Lab Program challenge with the financial support of this last award.

(Case 3) Cowlines App launched in several cities in Canada (ex: Toronto), and the USA (ex: NY, San Francisco, Los Angeles), integrating bike-share, car-share, ride-share, public transit, and taxis into a single customized route, facilitating citizens to move around the desired cities.

(Case 4) Safe & The City, launched in 2018, in London, to help people to reach their destination safely. It uses crowdsourced information, GPS, and Police Risk data to decrease the number of victims of opportunistic crimes.

(Case 5) The Wesolve—Better Together app was launched in March 2021 in Copenhagen to engage citizens and facilitate problem-solving and decision-making processes by using smartphones, challenges, surveys, polls, gamification, etc.

(Case 6) Toogethr Rideshare app developed in Amsterdam to make ridesharing with colleagues easier, more social, and more sustainable, contributing to the reduction of the carbon footprint, mobility cost reduction, and increased population satisfaction.

Official portal, website

An electronic portal/website is another digital technology widely used by the 25 benchmark SC to provide information on documents, products, and/or services to society.

They have been used in combination with several approaches to engage people over time, such as acts, advocacy, apps, ambassadors, articles, awards, apprenticeship programs, awareness campaigns, case studies, ceremonies, citizens juries, citizen advisory committees, civic crowdfunding, conference, congress, consensus building, challenges, citizen feedback/evaluation, citizen science, co-creation workshops, contests, database, dashboard, data visualization, deliberate poling, demo day, demonstration projects, design thinking, digital inclusion/literacy, fab/living/urban labs, finance incentive, focus group, funds, guidelines, hackathons, handbook, gamification, knowledge sharing, maps, open platforms, open innovation, panels, participatory budget, policies, projects, PPP, public kiosks, scholarships, showcases, smart community network, smart stories, social media, storytelling, survey, user-centered design, volunteering, etc.

A good example is the Amsterdam SC Platform (https://amsterdamsmartcity.com/), an open innovation platform where active citizens, companies, government, and knowledge organizations come together, collaborate and interact for the development of a green, smart, and healthy future of the Amsterdam Metropolitan Area.

Anyone can create an account, learn about news, events, and opportunities, share experiences and projects, and request partnerships or support to implement ideas/projects related to the Amsterdam SC Managers’ priorities: citizens & living, circular economy, digital city, energy, mobility, and SC Academy.

Thus, the Amsterdam SC Platform has twenty-eight permanent partners from the government, and from the knowledge, social, and creative industries in the Amsterdam Metropolitan Area. Its community of over 8,000 inventors implements projects like:

(Project 1) CityFlows: To enhance the livability of crowded pedestrian areas by providing decision support tools to manage pedestrian traffic flows.

(Project 2) CitySDK: a system to collect open data of the government, to provide its availability in real-time.

(Project 3) CIVIC: to find innovative solutions for construction logistics.

(Project 4) Digital Society School: to work with governments, businesses, and residents to help them adapt and become future proof for the digital world.

(Project 5) EMPOWER 2.0: to Empower the Citizen—Towards the European Energy Market 2.0.

(Project 6) Klup: to reduce loneliness by connecting seniors.

(Project 7) Re-Store: to evaluate and impact new solutions to process organic waste.

(Project 8) SC Kit: to permit the active involvement of common citizens to measure the quality of their air.

(Project 9) Smart Kid Lab: for children to map their environment playfully, by using modern technology and instruments.

(Project 10) The SC Lab: a workplace where Amsterdam SC partners meet and work together and lectures, workshops, open houses, and delegation visits are hosted.

(Project 11) The Hackable City: to explore the potential of new models of collaborative city-making in a network society.

(Project 12) Together: to share rides with colleagues easily by automatically providing the best matches and rewarding users through earning points. This project won The Hague Innovators 2017 public prize.

(Project 13) Roboat: to explore and test autonomous systems on water: deliver goods, collect waste, dynamic infrastructure, environmental sensing, and transport people.

Another official platform developed by the local authority of Amsterdam is New Amsterdam Climate (https://www.nieuwamsterdamsklimaat.nl/), as a result of initiatives started in 2019 to invite and dialogue with citizens, companies, universities, government, and other stakeholders to develop the Phase 1 of the Amsterdam Climate Neutral 2050 Roadmap, to reach the ambitious goal to reduce CO2 emissions in Amsterdam by 55% in 2030 and by 95% in 2050. Nowadays, the platform cataloged 386 projects in the city, developed by residents, companies, and other institutions for a healthy and sustainable city.

Open data platform

Official portals or websites are not exclusively dedicated to SC but also can be used as open data platforms, systems, or technology infrastructure designed to collect, process, and share a large amount of data openly and transparently.

The main goal of an open data platform is to promote access to government data and encourage the development of creative applications and tools to engage and serve the wider community (Martín, De Rosário & Pérez, 2015).

The widespread use of open data platforms around the globe can be seen through the 204 Smart Cities listed by RList Insights (2019), most located in North America (81; 39.7%; 68 in the USA and 13 located in Canada), followed by Europe (65; 31.8%; 13 in Italy, 12 in Spain, while France, Germany, and the UK have seven cites each), and Asia (40; 19.6%; with Japan leading with 14 cities, SK with nine cities and Taiwan with seven cities), with Amsterdam, London, NY, San Francisco, Singapore, Seoul, Paris, Shanghai, Tokyo, and Toronto considered the world’s ten top cities with well-designed city open data portals with rich datasets.

To reinforce part of the RList Insights (2019) findings, when the open data of 30 global cities was evaluated by The Digital Cities Index 2022 (The Economist Group, 2022, p. 23), it was found that European and North American cities dominated the open data access and use policies, covering the usage and publishing of data for accountability, innovation, and social impact, with emphasis on cities like London, Toronto, Paris, Dallas, NY, Washington DC, and Seoul.

When the 25-benchmark SC are investigated, it was found that all have developed an open data platform or portal, whose names and links are provided in Appendix 5. Although a detailed analysis was not carried out in each portal to check how many users were engaged, it is possible to trace a few commentaries about London and Toronto.

(a) London DataStore (https://data.london.gov.uk/), developed in 2010, now has 1,124 datasets organized in 18 topics, from Demographics (215) to London 2012 (6), available in 27 formats, including spreadsheet (669), PDF file (326), CSV file (250), Website (138), ZIP file (107), and GeoPackage (29). Furthermore, Transport for London (TfL) developed another public open data portal (https://tfl.gov.uk/info-for/open-data-users/) which has been used by over 5,000 developers, used in over 600 apps, and generated savings and economic benefits up to £130 m a year for TfL, London, and travelers (Deloitte, 2017 p. 5 to 10).

As a result, the Greater London Authority received in 2015 the ODI Annual Open Data Awards for opening a range of public sector data for the use of public, city staff, commercial organizations, and other public bodies.

(b) The Toronto Open Data (https://open.toronto.ca/). The city of Toronto launched its open data portal in 2009 to meet increasing demand for open data access. This was followed by the implementation of an open data policy in 2012, the establishment of a public sector open data working group in 2015, and ongoing portal development from 2017 onward.

The most interesting feature of this platform is the community engagement over time, for instance, in January 2018, the Open Data team developed the Open Data Master Plan & 4-Year Roadmap, co-developed with the community, with four themes (Foundation, Integration, Connection, and Activation) and twelve actions starting from the update of publication pipeline and ending to increase awareness of open data.

One example of public engagement was a recent questionnaire asking the public about the type of data they want to prevent, mitigate, and address issues related to the city’s five main priorities: affordable housing, climate change, fiscal responsibility, and mobility, in which the results are planned to be shared on the portal.

The Toronto Open Portal’s Data Catalogue has over 700 datasets in twenty topics, with the majority linked to locations and mapping (149), city government (139), community services (87), transit (86), and public safety (82).

The platform’s 212 Civic Issue datasets focus on mobility (65), poverty reduction (55), affordable housing (38), fiscal responsibility (29), and climate change (25). The portal offers data in 21 formats, including CSV (229), SHP (165), XLSX (150), JSON (129), and GEOJSON (113).

The Toronto Open Data team has also developed: (a) explanations about the datasets, project, and technical resources; (b) a knowledge centre with news published biweekly; (c) partnership with educators, schools, and students to help them use the data for their final projects; (d) a gallery with ten apps developed such as Cycle Now, Garbage Day, Recycle Wizard, Toronto API, etc.

Results and discussion of the two diagnostics and the ce kit model

Contextualization and first diagnosis (consulting the City Hall Managers)

In 2016, Mr. Arthur Virgílio Neto, the mayor of Manaus City, was reelected after a strong marketing campaign promising in his government plan, a project to transform Manaus into a SC. However, from 2016 until March 2023, only nine decrees were published by the Manaus City Hall quoting the term “Cidade Inteligente=SC” which are as follows:

(Publication 1) Decree 4,276, 3 January 2018, page 53: published by SEMEF (Finance Secretary) proposed a budget of R$ 22.3 million to implement an Information Technology Infrastructure, but only that, no information about the plan, activities, schedules, etc.

(Publication 2) Decree 4,357, 7 May 2018, page 9: published by SEMAD (Administration Secretary) authorization to Mr. Sérgio Augusto Magalhães de Souza, Division Chief of SEMEF, to participate in the Smart City 2018.

(Publication 3) Decree 4,386, 20 June 2018, page 10: a Minute published by Manaus PPP Committee where the company Fiscal Tech proposes to SC, the use of (1) an Urban Mobility System; (2) a Public Safety System with Cameras; (3) Support Systems and Multiservice Network with Fiber Optics. The Manaus PPP Committee authorized the company to realize technical studies on the subject.

(Publication 4) Decree 4,460, 10 October 2018, page 2: Manaus Strategic Planning 2030 by the mayor, which mentions “Smart City” once. Before this decree, on 17 July 2018, the mayor (https://semad.manaus.am.gov.br/?p=2541) launched the book “Manaus Strategic Planning 2030” in the auditorium of Amazon Industry Federation, but the term “Smart City” is vague, appearing only once on page 33 “Implementation of Smart City,” with no definition, goals, budget, schedule, or detailed plan.

(Publication 5) Decree 4,952, 22 October 2020, page 20: a public call published by SEMTEPI (Secretary of Job, Entrepreneurship, and Innovation), to select a Civil Society Organization to develop technological and innovation activities in the Casarão da Inovação Cassina, a Center for Entrepreneurship and Innovation. The term “Smart City” is cited twice, in the first, the SEMTEPI declares that the “Casarão da Inovação Cassina” will insert Manaus among the Brazilian Smart Cities, and the second just defines what is Smart City.

(Publication 6) Decree 4,956, 28 October 2020, page 14: the public call published by SEMTEPI (Secretary of Job, Entrepreneurship, and Innovation) by Decree 4,952, 22 October 2020 is repeated.

(Publication 7) Decree 5,155, 04 August 2020, page 8: published by SEMAD, granting per diems to Mr. Clênio Francine (Project Manager) and Sandro Elias (Secretary) to visit Instituto da Cidade Inteligente in Curitiba city.

(Publication 8) Decree 5,177, 03 September 2021, page 1: Mayor (elected in 2020) establishes Manaus 4.0 Technical Commission to implement Smart City concepts. It has 14 members from seven Manaus City Hall organizations. It must also generate a preliminary diagnostic of Manaus reality by December 31, 2021, and propose a plan of investments to adopt technology solutions to assist digital integration. However, Manaus residents don't know if those jobs have been done yet.

(Publication 9) Decree 5,178, 06 September 2021, page 01: public call published by SEMTEPI (Secretary of Job, Entrepreneurship, and Innovation) inviting civil organized society to participate in the project “More Innovation” to foster the innovation environment in Manaus. The term “Smart City” is cited only once “In an industrial city, workers cannot be lacking, as well as in a smart city, programmers, software developers, and entrepreneurs capable of creating innovative solutions for the daily use of the city cannot be lacking”.

In addition, to know more information about the Implementation of the Manaus SC Project, a questionnaire with seventeen open questions (Appendix 1) was sent to the Manaus City Hall Managers on May 23, 2022, and it was scheduled to be answered until 12 June 2022, but it took 268 days (almost nine months; 15 February 2023) for the researcher to receive the following answer from someone (Code 63322757234) located at SEMEF (Appendix 6): “Dear, we inform that Professor Jonas Gomes da Silva attended this Secretariat in the same period of 2022, in a meeting scheduled with the expense order of this Secretary, where he addressed all questions personally. As a result, I return the records for appropriate measures.”

Surprisingly, the researcher was never invited by the managers of the Manaus City Hall, nor did visit or engage in dialogue with the SEMEF Secretariat. As a result, after 6 years, based on the document analysis and false feedback received, the SC Project in Manaus felt like a political fallacy. No plan, project, or model, particularly with a citizen participatory approach, has been effectively adopted in the city of Manaus.

The second diagnostic (consulting Manaus citizens)

A total of 1,242 people responded correctly to this questionnaire, and the main results are resumed below:

(Gender) 655 (52.7%) identified themselves as females, while 580 (46.7%) as male, and seven (0.6%) preferred do not disclose their gender.

(Age) the youngest (18–24 years old) (393; 32%) were the majority, followed by the oldest (>50 years old) (287; 23%), perhaps due to higher time availability. A total of 213 (17%) middle-aged persons (40–50 years old) contributed, followed by 187 (15%) and 162 (13%) 25–30-year-olds.

(Education) 489 (39.4%) have superior education, 414 (33.3%) have high school education, 187 (15%) have specialization Lato Sensu, 79 (6.4%) have a Master Course, 54 (4.3%) have a Doctor Course, and 8 (0.6%) have fundamental education. Eleven respondents reported further levels.

When asked about their level of knowledge on GovTech, SC, and Decarbonized Cities, it was found that:

(a) Govtech is the most unknown term for 820 respondents (out of 1,193; 68.7%), while only 17 people (1.4%) declared that they have a high level of knowledge on the subject.

(b) Decarbonized cities is the second unknown term, with 468 people (43.1% of 1085) not knowing about it and 43 (4%) saying they know a lot about it.

(c) SC term is completely unknown for 305 (out of 1,118; 27.3%) respondents, while 82 (7.3%) informed that have a higher of knowledge about the term.

When asked how many times they had been invited by the public administration to participate in the construction of a sustainable plan for the Amazon State or Manaus City since 1988 (the year of Brazil’s re-democratization), the overwhelming majority of respondents reported that they had never been invited by the Brazilian Federal Government (1,116; 95.2%), Amazon State Government (1,047; 93.2%), or Manaus City Hall (1,038; 93.4%).

These results reveal that the public administration, from national to local, has not engaged citizens in the planning process to solve urban difficulties in Manaus city and likely in many Brazilian cities. Cultural factors may explain this, including a centralized old administration, lack of modernization, low transparency, and bureaucratic barriers due to corruption, lack of political will, low leadership, and resistance to change traditional planning and decision-making methods.

When each responder was asked, “If you were invited to help build a long-term plan to transform Manaus into a SC, would you participate?” 826 (66.5%) indicated they would participate, 383 (30.8%) answered maybe, and 33 (2.7%) said no. This indicates that most are willing to help. Awareness campaigns and clear communication may change the minds of those who are unsure.

Among the 12 service areas of the Manaus City Hall, the five main areas in which respondents would like to help transform Manaus into a SC are Environment (651; 52%), Education (580; 47%), Mobility (390; 31%), Health (363; 29%), and Governance (307; 25%).

As part of the survey, respondents were given 17 environmental issues and asked to choose the five most problematic in Manaus. Thus, the top five environmental issues are: (1) increased stream/river pollution (974; 78.4%); (2) increased street garbage (763; 61.4%); (3) insufficient urban afforestation, including trees without maintenance and urban development without adequate afforestation (612; 49.3%); (4) increased air pollution, including odor and stench (501; 40.3%); and (5) increased traffic congestion (493; 40%).

This result can provide valuable insights for policymakers and urban planners in Manaus. Addressing these environmental problems could involve measures such as improving waste management, promoting green infrastructure and sustainable urban design, and implementing policies to reduce pollution and traffic congestion.

Six courses related to SC, Decarbonized Cities, and DT were provided to respondents at different levels. They selected the type of training they would prefer to receive. The results showed that most respondents (772; 62.2%) preferred a short course on Smart or Decarbonized Cities, followed by a basic course on DT such as AI, Blockchain, and IoT (432; 35%), an advanced course on disruptive technologies such as AI, Blockchain, and IoT (305; 25%), a specialization lato senso course on Smart or Decarbonized Cities (291; 23.4%), a master course on Smart or Decarbonized Cities (228; 18.4%), and only 143 people (11.5%) wished to participate in a Doctoral course on Smart or Decarbonized Cities.

This presents an opportunity for educational institutions and training providers to develop courses and programs that cater to this demand.

Proposed participatory SC model (CE KIT)

Goal and importance of the model

The main goal of the CE KIT model is to transform citizens into protagonist during the SC journey. And this is important for several reasons, such as co-creation, empowerment, ownership, and sustainability.

Co-creation: Policymakers should use citizens’ diverse experiences and knowledge. Citizens should be recognized as protagonists in identifying urban problems and offering solutions rather than just end users of SC technologies. Citizen co-creation gives city planners and innovators insights into society needs and preferences, enabling more effective and user-centric solutions.

Empowerment: Assigning residents an active role in the SC process empowers them to contribute to decision-making and community development. This fosters greater engagement, motivation, and satisfaction.

Ownership: Engaging citizens throughout the SC journey cultivates a sense of ownership and responsibility for their communities. This can promote accountability, inclusiveness, civic pride, and willingness to maintain and improve SC solutions.

Sustainability: SC encompass more than just technology. Truly sustainable, livable, resilient communities emerge by involving inhabitants across SC initiatives, ensuring solutions align with the vision, needs, and values of the populace. This facilitates comprehensive social, economic, and environmental sustainability.

Some precautions before or during the use of the CE KIT model

Since the CE KIT model (Fig. 8) is a subset of the P5 model (Fig. 6) outlined in “P5 Model with Enablers that are Contributing to the Popularization of SC”, it is vital to ensure that the basic enablers are included in the foundation of the SC journey, especially:

Figure 8 Citizen engagement KIT model.

Leadership: Successful SC development requires effective leaders who can inspire stakeholders, develop a shared long-term Vision, create strategic plans, build partnerships, coordinate activities, and drive innovation.

Long-term vision: Decision-makers must comprehend the SC concept to create a long-term vision that meets citizen requirements. Effective leadership is needed to articulate an inspiring, motivating Vision that guides decisions and resource allocation across time.

Principles: Establishing guiding principles and values is vital for SC initiatives, as they shape daily decision-making, build trust, ensure inclusion and equity, and encourage innovation by providing a framework for standards, incentives, and experimentation.

Diagnosis: Data-driven diagnosis of a city’s opportunities and challenges is important for building stakeholder consensus based on facts, measuring progress, and optimizing resource allocation.

Goals and priority areas: Aligning goals and priority areas with the long-term vision enables effective resource targeting, measurable progress, engaged stakeholders, and strategic roadmaps for project development over time. Goals often relate to efficiency, sustainability, and QoL.

Budgets and incentives: Making budgets and incentives available is crucial for resourcing the smart city journey, stimulating participation through accountable, transparent funding allocation that drives efficient, effective engagement and investment.

ICT Infrastructure: Robust ICT infrastructure is essential for enabling automation, connectivity, data use, testing, and innovation across SC solutions.

When the above enablers are built as a foundation of a SC journey, then it is possible to explore the full potential of the CE KIT Model.

Characteristics of the CE KIT model

The CE KIT model is practical, diverse, flexible and has four parts, as described below.

(First) Practical: Based on the real cases learned from the best SC that developed formal models/policies to engage their citizens over time, such as Amsterdam, Melbourne, Montreal, San Francisco, Seoul, and Taipei.

(Second) Diverse: A wide range of approaches (approximately 120) were used by 25 Benchmark SC and other SC found in the literature and documentary review to involve residents.

(Third) Flexible: It enables policymakers and/or decision-makers to investigate the best cities’ models and approaches, study them, and select which one is more suited to their reality, or possibly blend them to create a new way to engage residents based on local needs.

(Fourth) Composed of four parts: PDSAR Cycle, IAP2 Spectrum, Formal Models developed in SC to engage residents, and Living Labs.

Part 1 of the Model: PDSAR Cycle

The first part is the PDSAR Cycle, adapted from the Shewhart PDCA and Deming PDSA cycles (Deming, 1982). This organizational learning approach has been utilized since the early 20th century and remains in use across manufacturing and service sectors to systematically solve problems and continuously improve product/service quality over time.

At the core of the PDSAR cycle should be the SC’s long-term vision or a specific goal. This methodology aims to better organize citizen engagement approaches over time, where:

P represents the planning phase, involving detailed plans for projects that may encompass (a) identifying the problem and potential SC solutions/technologies (e.g., sensors, IoT, cloud computing, big data analytics); (b) defining key performance indicators to measure success; (c) establishing implementation schedules and milestones; (d) allocating budget/personnel resources; (e) developing communications plans to update stakeholders, etc.

The P phase could also represent a policy, program, project, platform, partnership, or product to achieve the vision or goal. This phase offers 23 potential approaches for CE.

For example, the Appendix 7 contains a public challenge with a list of 12 basic project proposed in February 2023 by engineering students at the Federal University of Amazonas (UFAM). This document was developed by the author during quality management coursework.

To realize this public call, it was used an A3 report template model based on Toyota’s approach, a community evaluation sheet (for paper or app use), results, photos, and the 12 student team proposals. Projects aimed to address Manaus’ top environmental issues identified in this article’s “The Second Diagnostic (consulting Manaus citizens)”: (1) increased pollution of streams/rivers (974; 78.4%); (2) increased street garbage accumulation (763; 61.4%); and (3) insufficient urban reforestation—untended trees and inadequate greenery during development (612; 49.3%).

Approximately 60 participants properly evaluated the projects. The winning team was the team Manaus Arbor. This simulated public call could be conducted by Manaus City Hall, challenging students to propose basic 2024 projects for local communities within a R$150,000 budget. Key approaches used in this planning phase included: brainstorming, cases, diagnostics, community surveys, Ishikawa diagrams, rich pictures, challenge/goal trees, A3 reports, and 5W2H analysis.

The second phase is D (Develop), involving implementation of the P component (Program, Project, Platform, Plan, Partnership, or Product). Approximately 24 approaches can engage citizens during development, including alliances, apps, awareness campaigns, civic crowdfunding, etc.

The third PDSAR phase is S (Study), collecting and analyzing data from implemented activities to evaluate effectiveness in achieving defined objectives. Using the key performance indicators established during planning is recommended to measure success. This phase should identify any gaps, issues, areas for improvement, and document best practices or lessons learned for the next phase. Fourteen proposed approaches for CE during study include citizen feedback systems, dashboards, data visualization, gamification, interviews, open data platforms, etc.

The fourth phase is A (Act) based on findings from the “Study” phase. Two main actions are possible-first, implementing corrective actions to address any identified issues or gaps. Second, documenting and disseminating best practices for recognition. Approximately 20 proposed engagement approaches in this phase include advocacy, case studies, conferences, demonstrations, documentation, guidelines, handbooks, standardization, storytelling, showcases, etc.

The final phase is R (Reward), recognizing individuals/teams and celebrating results, which builds ownership, community engagement, trust, transparency, learning/sharing, and motivation. Sixteen recommended approaches include awards programs, apprenticeships, certifications, ceremonies, events, funds, innovation prizes, mentorships, etc.

The PDSAR cycle repeats after a goal is achieved by establishing a new one and continuing rotation, ensuring the model’s sustainability through improvements and adaptation to deal with new SC issues and opportunities.

Part 2 of the model—IAP2 Spectrum

If policy/decision-makers opt not to use the PDSAR methodology, the CE KIT model alternatively proposes the IAP2 Public Participation Spectrum (IAP2 International Federation, 2018) with five phases:

Inform: Provide public with information to understand issues, alternatives, and solutions. Seventeen approaches include chatbots, platforms, help desks, newsletters, mobile apps, etc.

Consult: Obtain public feedback on analyses, alternatives, and decisions. Keep citizens informed, listen to concerns, and provide feedback on how inputs influence choices. Ten approaches include apps, panels, workshops, research, focus groups, participatory budgets, etc.

Involve: Work directly with residents to ensure aspirations and concerns are understood and considered. Fourteen approaches include assemblies, challenges, citizen science, living labs, etc.

Collaborate: Partner with citizens throughout decision-making, from developing alternatives to selecting solutions. Thirteen proposed approaches include committees, engagement, consensus building, labs, design thinking, demos, training, incentives, contests, etc.

Empower: Place final decision-making in citizens’ hands, enabling them to implement solutions. Empowerment approaches include associations, cooperatives, ambassadors, juries, education, digital inclusion, social innovation, PPPs, participatory budgeting, etc.

Part 3 of the model—formal models developed in SC to engage citizens.

The third part of the CE KIT Model is focused on the formal models identified in six SC to engage citizens, which is explained below:

Amsterdam models: Capra (2014)’s thesis explains the history of the Amsterdam SC Program, its governance structure, and the various typologies used to engage residents. Henriquez, De & Kresin (2015, p. 28) present the seven steps used by the Amsterdam Smart Citizen Lab to engage citizens during the process to develop software—(Step 1) Meet (open invitation. people sign up and meet at a safe and neutral space); (Step 2) Match (encourage people to form groups based on shared interests, experience or commitment level); (Step 3) Map (help people to understand in more detail the problems/opportunities); (Step 4) Make (encourage people to develop solutions, for example by using open source software and hardware); (Step 5) Measure (test the solutions); (Step 6) Master (analyze the data); (Step 7) Mobilize (citizens, public authorities to take action on the findings).

Another initiative developed more recently was led by Amsterdam City Council with the citizens, the Climate Neutral 2050 Campaign, followed by four strategic steps: (1) Invitation to the city; (Step 2) Dialogue with the City; (3) Develop the Roadmap; (4) Implementation of the Roadmap.

Melbourne Model: The City of Melbourne (2021b), in consultation with the community, council, and employees, developed The Community Engagement Policy, probably the unique policy of this nature among the SC investigated. This Policy also uses The IAP2 Public Participation Spectrum, explained above. The policy details the principles that guide their work to deliver results through shared problem-solving, open dialogue, and meaningful participation.

In addition, since 2021, the City of Melbourne developed the Neighborhood Portal and Model (https://participate.melbourne.vic.gov.au/neighbourhoods), in which planning process for the community experience involves listening, exploration and realization processes.

Montreal model: in the Montreal Smart and Digital City (2014–2017 Strategy), it explains that the model used by and for citizens to develop a Strategy composed of five steps:

(Step 1) Formulation of the city vision; (Step 2) involves listening through surveys, consulting residents, city workers, as well as investigating best Smart Cities practices, from international experience, identifying needs, issues, and priorities; (Step 3) defining strategic operations, selecting criteria and seeking approval from the city decision making bodies; (Step 4) development of an action plan, prioritizing short-term projects, major projects, and seeking approval from decision-making bodies; (Step 5) implement and follow up, deploying initiatives, ongoing reviews and evaluating the KPIs (Ville de Montréal, 2015 p. 10–41).

San Francisco model: this is an interesting case involving mobility, and San Francisco was among the cities selected in 2016 by the U.S. Department of Transportation National SC Challenge.

According to the San Francisco Municipal Transportation Agency (2018, p 8–10), the Community Engagement Plan has six goals and explores SC problem solving via the community challenge, where the community upload problems electronically, vote/comment on problems, communities form groups around a specific problem, and groups prepare and submit a proposal.

In this methodology, they encourage citizens to submit problems, assist them with popular ideas to form groups, and assist groups during the process of creating the applications, by using focus groups, public awareness campaigns, and baseline surveys.

Seoul model: according to Lee (2021 p. 12), the process used by the Seoul Digital Foundation to engage citizens has four stages, from urban problem diagnosis to specific solution—(Stage 1) discover (divergence) by using design thinking, workshop, and joint research; (State 2) define (convergence) by using education, digital literacy, maker space, and digital citizenship; (Stage 3) develop (divergence) by prototyping, testbed, living lab, maker space, and seminars; (Stage 4) deliver (convergence) by sharing knowledge, exploring Hackathon, Festival, Digital Urban Policy, and Makers Faire.

Taipei model: this city is a good case of a public-private partnership. Leu et al. (2021) and Taipei City Government (n.d.) informed that Taipei City established the Taipei SC Project Management Office (TPMO) in 2016, exploring the slogan “Government as a Platform, City as a Living Lab”.

TPMO supports the opening of the public test field and the introduction of creativity and resources from the collaborative private sector to promote top-down (for immature solutions) and bottom-up proof of concept (PoC) projects. The model proposes problems set by the government (public call for proposals), and problem-solving by the industry by using the top-down approach.

The Taipei SC Industrial Empirical Proof of Concept Program is a bottom-up approach to solve local problems involving themes with mature solutions. Although it is not clear how the model actively engages common residents, this model can be useful for those public decision-makers interested to know how to balance top-down and bottom-up approaches toward PPPs.

Finally, the fourth part of the CE KIT Model is dedicated to Living Labs as an effective way to engage citizens, with cases from Amsterdam, Singapore, London, Barcelona, and Montreal.

Conclusions, limitations and recommendations

This extensive research aimed to address five key questions (Q). Based on data collection and analysis, the following conclusions (C), limitations (L), and recommendations (R) can be drawn:

Q1. Given climate change and population growth, how can we protect future generations?

C1.1 Improve people’s life using historical authors’ wisdom. Charles Mulford Robinson stressed civic beauty in city planning to uplift the spirit of citizens. Frederick Law Olmsted, the first to respond to a query on Intelligent City, argued for urban public green areas for citizen well-being and egalitarian access. Patrick Geddes viewed cities as organisms rather than mechanical systems and advocated for a holistic, citizen-centric, and scientific approach to urban planning.

These writers highlight thoughtful, inclusive, and dynamic urban planning with active public engagement and multidisciplinary problem-solving.

C1.2 Future generations are at risk if we do not solve environmental challenges and shift to a low-carbon economy. The transition requires active citizen participation, adequate financing, modernized state, an open and smart government, visionary leadership, transparent policies, collaboration, investment in R&D, innovative solutions, responsible use of disruptive technologies, and updated regulations.

L1. This research only focuses in finding models and approaches to engage citizens in SC, not focused on initiatives developed by Eco, Green, Net0, Carbon Neutral or Sustainable Cities.

R1.1 New studies should examine (a) which approaches found in this study are more effective to engage residents during the transition to a low-carbon economy; (b) the most effective financing models to support decarbonization initiatives; (c) the application of DT in renewable energy sources, carbon capture technologies, smart grids, EVs, etc.

R1.2 To adopt DT to enhance carbon measurement processes in the Amazon region could significantly improve emissions monitoring and support decarbonization efforts.

R1.3 To develop a methodology to decarbonize Brazilian Amazon Region. A system that incorporates sustainable solutions across the Energy, Land Use, and Forestry sectors through a collaborative multi-stakeholder process may offer the most effective path to decarbonization.

Q2. Does Manaus City Hall SC project work?

C2. No. The SC Project announced by Manaus City Hall Mayors since 2016 has not materialized. After 6 years, it feels like a political fallacy, there is no evidence of project, plans, programs, or models taking a citizen-participatory approach.

L2. The study did not examine why Manaus City Hall did not adopt the Project announced in 2016.

R2.1 Manaus City Hall’s decision not to adopt the SC Project should be investigated.

R2.2 Manaus has an opportunity to leverage its abundant natural resources, academic institutes, and private sector presence to establish itself as a Living lab for emerging technologies. Specifically, policy makers and researchers should consider (a) tapping into solar, hydroelectric, and other renewable energy sources to power a smart electricity grid; (b) engaging Manaus’ 600 industrial park companies with local universities and research centers on hydrogen applications, solar panels, sensors, electric vehicles, and sophisticated communication networks pilots.

Q3. How do Manaus residents view SC, decarbonization, DT, and environmental issues?

C3.1. There is a concerning lack of public knowledge regarding Govtech, Decarbonized Cities, and SC in Manaus.

R3.1 Manaus City Hall should invest in awareness campaigns, leveraging social media, events, workshops, and grassroots engagement. Collaboration with academia, NGOs, and other stakeholders could facilitate educational programs and capacity building to empower people.

C3.2 This study indicates a strong demand in training on Smart, Decarbonized Cities and/or DT like AI, blockchain, and IoT. Most respondents selected short courses, more convenient for workers. In the context of Smart and Decarbonized Cities, the interest in basic and advanced DT courses demonstrates that people understand the need of staying updated.

L3.2 This study did not analyze how to make individuals smarter, a crucial gap for future research.

R3.2 It is highly recommended to use an interdisciplinary scientific approach to develop an effective education and training system to prepare the citizen to address main city urban problems over time, further research or policy measures should be done to fill this gap.

C3.3 While Brazilian law mandates democratic, participatory urban governance, Manaus exhibits a stark lack of resident’s involvement in public administration initiatives. Over 93% of respondents reported never being invited to contribute to the Manaus SC Project or sustainable development plans, despite constitutional rights and statutes.

L3.3 and R3.3.1 This study did not investigate effective communication strategies to inform, sensitize and engage citizens, which open opportunities for new research.

R3.3.2 Brazilian policymakers and decision makers should modernize the state. Measures should include improving governance transparency, official communications on public involvement opportunities, participatory decision-making, and facilitating grassroots community organization. By embracing participatory SC principles, and empowering citizens to help shape Manaus’ future, the city can align with its legal foundations and tap into its greatest resource—its people.

C3.4 Most (66.5%) Manaus respondents are eager to help transform Manaus into a SC in key areas like Environment, Education, Mobility, Health, and Governance. To capitalize on this supportive spirit, a strategic participatory model is needed to empower citizens as partners in co-creating context-specific smart solutions, making the proposed CE KIT an asset for the city. However, almost one-third (30.8%) were unsure about their desire to engage, suggesting that engaging the public may be challenging.

L3.4. and R3.4 It may be important to investigate the reasons behind this uncertainty, such as a lack of awareness or understanding of what a SC entails, concerns about the impact of such initiatives on the community, or a lack of trust in government institutions or politicians.

C3.5 The five main environmental issues are related to stream/river pollution, street rubbish, insufficient urban afforestation, air pollution (including odor and stench), and traffic congestion.

L3.5 This study did not examine how Smart or Sustainable Cities have effectively addressed these environmental issues.

R3.5.1 Future research should investigate specific solutions from Smart or Sustainable Cities.

R3.5.2 Based on the findings, Manaus City Hall administrators should prioritize these pressing urban environmental problems. Potential strategies could incorporate the CE KIT Model with public calls as emulated in the UFAM student workshops (Appendix 7).

Q4) Which SC have the most inspiring CE models?

C4.1 The term SC has outperformed others in publications. This prevalence signifies that its framework, far from perfect, has emerged as a leading model to address pressing urban challenges.

C4.2 Most of the 25 benchmark SC have defined SC and/or have set a Vision for their SC initiatives in formal documents or on digital platforms. Most of these cities have frameworks, plans, programs, projects, and/or roadmaps to build, implement, evaluate, and share best practices.

C4.3 However, of the 25 benchmark SC examined, only six (24%) had formal approaches for CE. Four cities (Melbourne, Montreal, San Francisco, and Taipei) used CE models developed by city hall managers, while the remaining two cities (Amsterdam and Seoul) utilized models created by a living lab and a foundation affiliated with city halls.

R4.3.1 This suggests residents’ participation is not being sufficiently or systematically incorporated in the International Ranking, and in SC initiatives for most cities. More effort is needed to develop official policies, procedures, and frameworks to enable robust CE in SC.

R4.3.2 International SC rankings should include public involvement metrics to evaluate SC efforts comprehensively. Therefore, further study is needed to propose variables or indications for these rankings.

C4.4 The CE KIT Model for Manaus was inspired by formal CE models discovered in the SC of Amsterdam, Melbourne, Montreal, San Francisco, Seoul, and Taipei.

R4.4 Several SC projects fail due to poor foundations. This research highlights the significance of an interdisciplinary discussion to produce a clear SC Definition, Vision, and CE Model in official, publicly accessible documents. Effective alternatives include acts, laws, strategies, policies, programs, projects, roadmaps, blueprints, guidelines, handbooks, and government websites. Cities may create a strategic roadmap, coordinate stakeholders, and raise citizen awareness by investing in rigorous preparatory planning and documentation.

Q5) How can Manaus’ challenges be addressed using a citizen-centric SC model?

C5. Though a flexible and practical model. While the P5 model and CE KIT model are not 100% perfect, their implementation has the potential to yield benefits for the city of Manaus:

Firstly, the P5 model can help city officials identify and prioritize key enablers for SC development, including leadership, long-term vision, principles, transparency, legal frameworks, innovation ecosystem, education and training, alliances, diagnoses, CE KIT, budget, incentives, ICT infrastructure, funds, technologies, management, best practices, and more. This can streamline the development process and ensure that initiatives are focused on the most critical areas.

Second, the CE KIT model provides two main methodologies, the PDSAC cycle and the IAP2 Public Participation Spectrum, through which approaches can be selected and implemented. However, it is necessary to establish participative governance and cultivate a culture of continuous learning as keyways to building sustainability and improvement. As technology, legislation, demands, and settings evolve, the P5 and CE KIT models will be able to adapt.

The CE KIT model is also composed by the best SC CE models and Living labs, with 120 different approaches to support the transformation of citizens into active participants. This is critical since citizens in SC remain relatively excluded (Dameri, 2013, p. 2545; Paskaleva, Evans & Watson, 2021, p. 397), while they should be seen as the main actors in SC policies and putting them at the center means co-constructing policies with them throughout the policy cycle (OCDE, 2020, p. 7).

However, there are limitations that open several opportunities for new studies:

L5.1 and R5.1 This study does not go into detail on each approach stated in the CE KIT model, so more research should be done on these approaches.

R.5.2 The CE KIT model emphasizes active citizen participation through living labs, data collection, and digital platforms. However, SC technologies and extensive data gathering also pose privacy risks like surveillance, profiling, and unauthorized data sharing.

Although cyber security is one approach of the ICT Infrastructure enabler (Fig. 6) and development phase of the PDSAC Cycle (Fig. 8), more research is needed to develop a privacy risk management framework tailored to the CE KIT model, to assess and mitigate privacy threats in citizen-centric SC programs/projects.

Further research also should be done to investigate how living labs and/or digital platforms are effective to engage residents towards decarbonization by 2030/2050.

L5.2 While this research is built based upon existing literature (e.g., Dameri, 2013; Capra, 2014; Russo, Rindone & Panuccio, 2014; Albino, Berardi & Dangelico, 2015; Ferrer, 2017; Fernandez-Anez, 2016; Fernandez-Anez, Fernández-Güell & Giffinger, 2018; Eremia, Toma & Sanduleac, 2017; Janik, Ryszko & Szafraniec, 2020; Belausteguigoitia et al., 2022), a comparative analysis or discussion with these and other authors was not undertaken for the following reasons: (a) the goal is not to critically evaluate or discuss other works, as this is beyond the general scope of this general literature review; (b) the specific questionnaires utilized here were not published in previous scholarly journals, preventing direct comparison of results; (c) including a detailed discussion will make this article lengthier.

R.5.2 Not discounting the merit of this type of discussion, the rationale for an explanatory, non-comparative methodology is valid given the goals, original research content, and scoping nature of this work. Future research could involve more extensive comparison with previous findings.

Appendix

Appendices 1–7 can be found at: https://doi.org/10.7910/DVN/RK2PPK

Supplemental Information

Supplemental Information 1 Questionnaire sent to Manaus City Hall Managers.

Click here for additional data file.

Supplemental Information 2 Questionnaire sent to Manaus citizens.

Click here for additional data file.

Supplemental Information 3 Raw Data and Statistics to select 25 Benchmark Cities.

Click here for additional data file.

Supplemental Information 4 Example of Enablers that support Smart Cities.

Click here for additional data file.

Supplemental Information 5 Profile of Twenty-Five Benchmark Smart Cities.

Click here for additional data file.

Supplemental Information 6 Response from the Manaus City Hall to the Questionnaire.

Click here for additional data file.

Supplemental Information 7 The Public Challenge developed by the author at UFAM to engage engineering students to propose basic projects to address Manausć main environmental problems.

Click here for additional data file.

Supplemental Information 8 Raw data of the Questionnaire.

Click here for additional data file.

Thanks to responders, UFAM students, editors, and the four reviewers for their substantial research efforts. I also appreciate Miss Chole Best’s English corrections.

Additional Information and Declarations

Competing Interests

Author Contributions

Data Availability

The author has no conflicting interests.

Jonas Gomes da Silva conceived and designed the experiments, performed the experiments, analyzed the data, prepared figures and/or tables, authored or reviewed drafts of the article, literature review and development of the model, and approved the final draft.

The following information was supplied regarding data availability:

The raw data are available in the Supplemental Files.

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
