# Peer review of "Guidelines for a participatory Smart City model to address Amazon’s urban environmental problems"

_PeerJ Computer Science, doi:10.7717/peerj-cs.1694_

## Round 0.1 · original submission · Minor Revisions

Dear author,

After reading all the reviews, please make the listed edits to your manuscript. Please adapt your work to all recommendations.

Reviewer 1 ·

Basic reporting

The topic is very interesting connected with the new concept of smart cities.
The language of the text is ok with only minor errors.
Literature references are very good. Authors based the paper on extensive amount of new good literature in the field.
The article structure is ok.
The formal results contains all material needed.

Experimental design

The topic of the paper is within Aims and Scope of the Journal.
The paper lacks explicitly described research gap. The research questions well prepared the same is with research goals. Please link the research goals and/or research questions with literature.
The description of the research methods are described in good, extensive way.

Validity of the findings

The paper lack discussion with other literature – Authors should add it as a whole section to the paper.
The limitations of the paper should be described.
Please write what is new in your paper how it differs from others similar papers existing in per reviewed journals.
Add some potential social implication of the paper.

Reviewer 2 ·

Basic reporting

The author has presented guidelines for the participatory Smart city model to address Amzaon's urban requirement. The author has selected a very apt and hot topic for the discussion; The author has painstakingly done the review work and presented the data very effectively. Overall, this paper puts many essential issues in Smart cities, environmental concerns, etc. The Strength of the article is 1. Detailed Related Work Report 2.Survey work and, 3 Raising the essential issues.

Experimental design

The entire Study has been conducted in 4 phases, 1. Literature Review, 2 Consulting with the City Hall managers., 3. Consulting Manaus Citizens, and 4 Proposed a participatory Smart City Model with the recommendation.

The Authors have conducted extensive research with the phases mentioned above. However, I suggest the following.

1. Smart Cities, with their cultural aspects, are also important. The author may consider adding People Satisfaction and happiness index.
2. Our general assumption about the new technology is that it will help humankind, but it is good to think/evaluate it from the Social impact perspective. (See the impact of mobile technology on humanity, it has made the person more individualistic )
3. The proposed model must consider the personal data security and sustainability of the model.

Validity of the findings

The author has done detailed work in the study. The reported results are convincing. However, its impact study may have added value to the paper.

Additional comments

No further comments.

·

Basic reporting

The English language written throughout research is proficient. Clearly used the reference in the abstract as the idea of the author is study how the smart city model is evolved over years. This study helps to define the best possible guidelines to the Manaus citizens. Author has taken year-old examples to showcase how smart cities are evolved in ages. The abstract word count is with in the expected limitation.
Introduction: The samples taken are years old. The comparison is between 250+ years. Author has performed intense study of the smart city establishment, infrastructure, nature support, resource collaboration, funds generation and cultural ethics and empathically written the introduction and the conclusion is ended with important findings and Ten recommendations to the Manaus community.

Experimental design

Author has performed the study within the purview of his scope of the subject. Author has spent lot of time and resources to meet his end goal. He has gone beyond the mile to understand the type of written stories from very old books written 1952 and older. I appreciate author capability in order to getting the older material for his study.

Validity of the findings

Author has well-articulated the survey methodology with relevant examples and that data is well utilized in defining the final recommendations.
Author has touched how the ancient city structure evolved into IoT smart city. The metrics published are up to the mark and clear.
The size of the analysis number is high and accurate. Community engagement model, targeted citi count and size. Who are contributing to the smart cities. Over years how many publications are existing on smart city concept.

Additional comments

Strength: Vast study of the concept to define top ten recommendations.
Weakness: Some of the older books are not digital hence online citation is not possible. However, those references are necessary to complete the study.

·

Basic reporting

1. The paper is written in clear English without grammar mistakes or typos.
2. Several figures are of low quality and need to be improved to meet the journal's publishing standards.
3. The reference section needs to be improved as I can find several issues. For example:
'''
Kaneko, Y. (2001). Promoting ‘Electronic Government’ with a focus on Statistical Activities". In: IAOS Satelite Meeting on Statistics for The Information Society. [online] Tokyo, p.18. Available at: https://www.stat.go.jp/english/info/meetings/iaos/pdf/kaneko.pdf
'''
in line 2338, where is the opening double quote?
4. The organization of the paper also needs to be improved. The main sections should be enumerated, otherwise, it is difficult for readers to follow. The section "Results and Discussion" is overly detailed and unnecessarily prolonged. I also suggest subsection "3. Literature, bibliometric, and documentary review" be a new section since it is not necessarily a result or finding.

Experimental design

The research questions and goals are clearly stated at the beginning of the paper. The methodology is well-defined in the paper and the authors take a systematic approach to design the model based on related literature.

Validity of the findings

The novelty of the proposed model is somewhat decent and can be used as a guiding model for citizen-centered city models. I just want to question: what are the limitations of the current model and possible future work?

Additional comments

Overall I enjoy reading the paper.

---

## Round 0.2 · Minor Revisions

Dear Author,

Thank you for submitting your manuscript to the PeerJ Journal. Please address the issues you have identified with Tables/Appendices and resubmit.

---

## Round 0.3 · Minor Revisions

I ask the author to redo the abstract. In this section, do not provide literature but substantively justify the importance of the manuscript topic, provide the purpose of the work, the methodology used, and the results achieved. Also, please pay attention to the literature. Remove old items from 1913 and refer to the latest studies.

---

## Round 0.4 · accepted · Accept

This manuscript is ready for publication.

·

Basic reporting

Compared to the previous version, the paper has been substantially improved and easier to follow.

Experimental design

no comment

Validity of the findings

no comment

Additional comments

no comment